# ON selectivity in the *Drosophila* visual system is a multisynaptic process involving both glutamatergic and GABAergic inhibition

Sebastian Molina-Obando[1,2,3], Juan Felipe Vargas-Fique[1,2,3], Miriam Henning[1,2], Burak Gür[1,2,3], T Moritz Schladt[4], Junaid Akhtar[1], Thomas K Berger[4,5], Marion Silies[1,2]*

[1]Institute of Developmental Biology and Neurobiology, Johannes Gutenberg-Universität Mainz, Mainz, Germany; [2]European Neuroscience Institute Göttingen – A Joint Initiative of the University Medical Center Göttingen and the Max-Planck-Society, Göttingen, Germany; [3]International Max Planck Research School and Göttingen Graduate School for Neurosciences, Biophysics, and Molecular Biosciences (GGNB) at the University of Göttingen, Göttingen, Germany; [4]Department of Molecular Sensory Systems, Center of Advanced European Studies and Research (caesar), Bonn, Germany; [5]Institute of Physiology and Pathophysiology, Philipps-Universität Marburg, Marburg, Germany

**Abstract** Sensory systems sequentially extract increasingly complex features. ON and OFF pathways, for example, encode increases or decreases of a stimulus from a common input. This ON/OFF pathway split is thought to occur at individual synaptic connections through a sign-inverting synapse in one of the pathways. Here, we show that ON selectivity is a multisynaptic process in the *Drosophila* visual system. A pharmacogenetics approach demonstrates that both glutamatergic inhibition through GluClα and GABAergic inhibition through Rdl mediate ON responses. Although neurons postsynaptic to the glutamatergic ON pathway input L1 lose all responses in *GluClα* mutants, they are resistant to a cell-type-specific loss of *GluClα*. This shows that ON selectivity is distributed across multiple synapses, and raises the possibility that cell-type-specific manipulations might reveal similar strategies in other sensory systems. Thus, sensory coding is more distributed than predicted by simple circuit motifs, allowing for robust neural processing.

DOI: https://doi.org/10.7554/eLife.49373.001

*For correspondence:
msilies@uni-mainz.de

**Competing interests:** The authors declare that no competing interests exist.

## Introduction

Animals rely on their sensory systems to process behaviorally relevant information. One common feature of sensory systems is the sequential processing of information to extract complex features from simple inputs. For example, in the visual system, photoreceptors detect light and then downstream neurons progressively extract distinct features, such as contrast, direction of motion, form, or specific objects (*Gollisch and Meister, 2010*; *Livingstone and Hubel, 1988*). Sensory pathways diverge into pathways that become selective for increasingly specific features.

One prominent example is the split into ON and OFF pathways, where individual neurons become selective to either increases (ON) or decreases (OFF) in a signal. Such an ON/OFF dichotomy enables more efficient coding of stimuli in the visual system (*Gjorgjieva et al., 2014*) and occurs

**eLife digest** We rely on our senses to capture information about the world around us. Sense organs convert sensory information – such as light or sound waves – into patterns of neuronal activity. In the mammalian retina, for example, specialized neurons called photoreceptors detect individual photons of light as they hit the back of the eye. The photoreceptors then pass on this information to neurons called bipolar cells for further processing.

During darkness, all photoreceptors release the same chemical signal onto bipolar cells, namely a molecule called glutamate. But bipolar cells respond to glutamate in different ways depending on which proteins are present in their outer membrane. So-called ON cells respond to glutamate by decreasing their activity, and thus effectively become more active when light levels increase. By contrast, OFF cells respond to glutamate by increasing their activity. This ON/OFF binary code enables later stages of the visual system to detect more complex visual features, such as shape and movement.

A new study in fruit flies, however, suggests that the ON/OFF code may be more complex than previously thought. While fruit fly eyes look very different to our own, the two have much in common. By studying fruit flies, researchers can also take advantage of a variety of genetic and pharmacological tools to manipulate cells and neuronal circuits.

Using such tools, Molina-Obando et al. show that the ON/OFF signal separation in fruit flies uses two different molecular mechanisms. The first involves a gene called GluCl-alpha, which encodes a receptor for glutamate. The second involves a gene called Rdl, which encodes a receptor for another brain chemical, GABA. Deleting the gene for GluCl-alpha from the entire fly brain prevented ON cells from responding to an increase in light levels. However, deleting this gene from specific ON cells alone did not. This suggests that flies can use more than one type of neuronal connection to detect an increase in light. Moreover, if one pathway fails, the other can take over. This makes the system more robust.

The results of Molina-Obando et al. are consistent with findings from anatomical studies that have mapped connections between neurons. Future studies should explore whether the same mechanisms exist in other sensory systems, and other animals. These experiments could take advantage of the molecular tools developed as part of the current work, which allow precise manipulation of neural networks.
DOI: https://doi.org/10.7554/eLife.49373.002

across many different species and sensory modalities, such as vision, olfaction, audition, thermosensation, and electrolocation (*Bennett, 1971*; *Gallio et al., 2011*; *Scholl et al., 2010*; *Tichy and Hellwig, 2018*; *Werblin and Dowling, 1969*). Examples of how the split into ON and OFF pathways is implemented in sensory information processing have already been described. In the vertebrate retina, ON and OFF pathways split downstream of glutamatergic photoreceptors where ionotropic glutamate receptors on OFF bipolar cells maintain the sign of the response in the OFF pathway, and the metabotropic glutamate receptor mGluR6, located on ON bipolar cells, inverts the sign in the ON pathway (*Koike et al., 2010*; *Masu et al., 1995*; *Vardi, 1998*). In the olfactory system of *C. elegans*, an odor response can be split into parallel pathways in which glutamate-gated chloride channels mediate the ON response (*Chalasani et al., 2007*). While these transformations are thought to occur at specific synapses, connectomics data reveals that neural circuits are intricate and that many of the possible neuronal connections are realized (*Eichler et al., 2017*; *Takemura et al., 2013*; *Zheng et al., 2018*). This argues that important signal transformations might actually be distributed across wider circuit motifs.

In the *Drosophila* visual system, ON and OFF pathways functionally split in the first order lamina interneurons, but the physiological specialization occurs one synaptic layer further downstream. In brief, information travels from the retina, which houses the photoreceptors, through three optic ganglia: the lamina, the medulla, and the lobula complex, comprising lobula and lobula plate (*Figure 1A*). Contrast is encoded by the transient response of photoreceptors, and downstream lamina neurons amplify the contrast-sensitive signal component (*Laughlin, 1989*). Then, distinct ON and OFF pathways are required to detect contrast increments and decrements, respectively

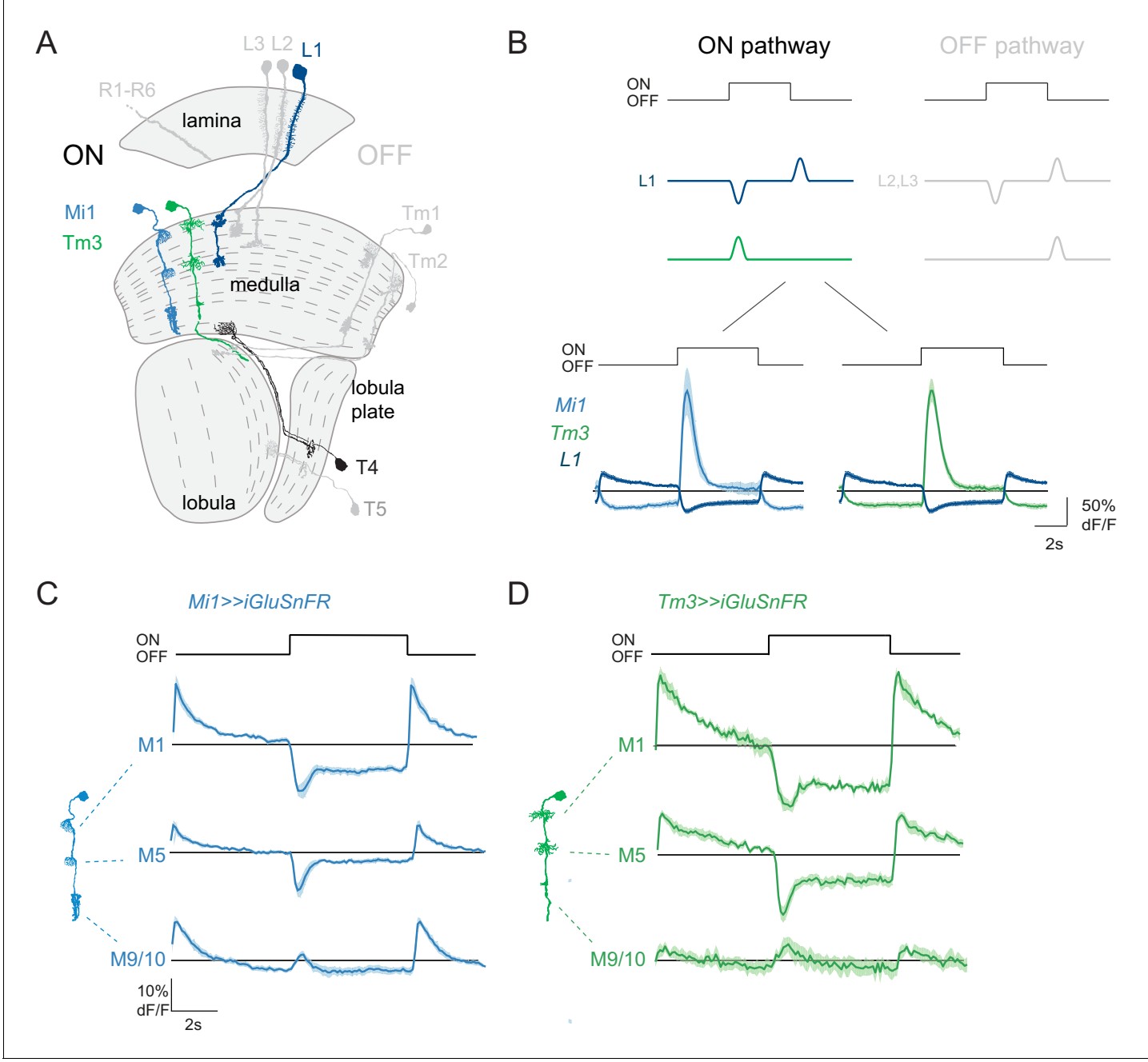

**Figure 1.** ON pathway medulla neurons that receive graded, glutamatergic input. (**A**) Schematic of the fly visual system, highlighting major neurons of the ON (colored) and OFF (gray) pathways. Visual information travels from the photoreceptors (R1–R6) through the lamina, medulla and lobula complex (lobula + lobula plate). In the ON pathway, the L1 input and its two major postsynaptic targets Mi1 and Tm3 are highlighted, as well as their common target, the T4 ON-direction-selective cell. (**B**) Top: Schematic representation of the signal transformations that occur at the lamina-to-medulla neuron synapse. In the ON pathway, a sign inversion is required downstream of linear lamina neuron inputs. Bottom: In vivo calcium signals in response to 5 s full-field flashes. L1 calcium signals (dark blue, n = 6 [99]) are of the opposite sign to the calcium signal in its major postsynaptic partners Mi1 (light blue, n = 5 [89]) and Tm3 (green, n = 7 [84]). (**C,D**) In vivo iGluSnFR signals in response to 5 s long full-field flashes at the dendrites of Mi1 (C, n = 9[278] in layer M1, n = 9[250] in M5) and Tm3 (D, n = 6[137] in M1, n = 6[141] in M5), or at the output region of these neuron types in medulla layers M9/10 (n = 9[326] for Mi1 in C, n = 6[161] for Tm3 in D). All traces show mean ± SEM. Sample sizes are given as number of flies [number of cells].

DOI: https://doi.org/10.7554/eLife.49373.003

(*Joesch et al., 2010*; *Strother et al., 2014*). In the lamina, L1 is the major input to the ON pathway, whereas L2 and L3 feed into the OFF pathway (*Clark et al., 2011*; *Joesch et al., 2010*; *Rister et al., 2007*; *Silies et al., 2013*). The assignment of L1, L2, and L3 to ON and OFF pathways originates from neuronal silencing studies (*Clark et al., 2011*; *Joesch et al., 2010*; *Silies et al., 2013*). However, all lamina neurons receiving direct input from photoreceptors depolarize to the offset of light and hyperpolarize to the onset of light (*Clark et al., 2011*; *Laughlin, 1989*; *Silies et al., 2013*; *Uusitalo et al., 1995*), thus passing on information about both ON and OFF (*Figure 1B*). Voltage or calcium signals in most downstream medulla neurons then selectively report only one type of contrast polarity. The major ON pathway medulla neurons Mi1 and Tm3, for example, selectively respond with depolarization or an increase in calcium signal to ON (*Figure 1B*; *Behnia et al., 2014*; *Strother et al., 2017*; *Yang et al., 2016*). In the OFF pathway, most neurons instead selectively respond to OFF stimuli, retaining the response polarity of their lamina inputs (*Figure 1B*; *Behnia et al., 2014*; *Serbe et al., 2016*; *Yang et al., 2016*). Therefore, ON selectivity requires a sign inversion between the L1 input and its postsynaptic partners Mi1 and Tm3. Previous work suggested that the L1 input to the ON pathway is glutamatergic, whereas L2 and L3, the two major inputs to the OFF pathway, are cholinergic (*Davis et al., 2018*; *Takemura et al., 2011*). This suggests that glutamate might also be used as an inhibitory neurotransmitter to implement ON/OFF dichotomy in the fly visual system. However, the molecular and cellular mechanisms implementing this signal transformation are not known in *Drosophila* visual circuitry.

Connectomics data has generated predictions about core circuit motifs (*Shinomiya et al., 2014*; *Takemura et al., 2013*). In the ON pathway, L1 makes the largest number of synapses with the medulla intrinsic Mi1 neuron and the transmedullary Tm3 interneurons (*Figure 1A*; *Takemura et al., 2013*). However, L1 has many other outputs, and Mi1 and Tm3 many additional inputs, such as indirect L1 input via L5, or the GABAergic neuron C2, among others (*Takemura et al., 2013*). Thus, coding in the visual system could be distributed across parallel pathways. One synaptic layer downstream, Mi1 and Tm3 medulla neurons project to T4, the first direction-selective cells of the ON pathway (*Figure 1A*; *Fisher et al., 2015a*; *Maisak et al., 2013*). This core visual circuit motif appears to be surprisingly resilient to perturbations. While genetic silencing of Mi1 or Tm3 leads to some deficits in ON edge motion detection (*Ammer et al., 2015*; *Strother et al., 2017*), these flies are not ON motion blind, arguing that other neurons must also play a role in motion detection. Mi4 and Mi9 have now been added to the ON pathway (*Takemura et al., 2017*). These cell types are modulated by octopamine, but silencing Mi4 or Mi9 individually has only subtle phenotypes (*Strother et al., 2018*; *Strother et al., 2017*). In the OFF pathway, combinatorial block of more than one cell type aggravates behavioral deficits (*Fisher et al., 2015a*; *Serbe et al., 2016*; *Silies et al., 2013*). This is also already true for the lamina neuron inputs L1, L2 and L3 neurons (*Silies et al., 2013*). This could argue that individual neurons might have distinct, but overlapping tuning properties (*Serbe et al., 2016*; *Tuthill et al., 2013*). Alternatively, encoding of a single aspect of a feature might already be distributed across parallel pathways.

Here, we show that ON selectivity in the *Drosophila* visual system is mediated by a glutamate-gated chloride channel, GluClα, and that all ON responses are lost upon pharmacological block or genetic loss of GluClα in the entire brain. At the same time, ON responses are robust to cell-type-specific perturbations of GluClα in individual neurons postsynaptic to L1, arguing for the existence of parallel functional pathways. Furthermore, we found that GABAergic inhibition also plays a role in mediating ON responses downstream of the glutamatergic L1 input. Together, our results indicate that ON selectivity is a multisynaptic computation that depends on both glutamatergic and GABAergic inhibition. This suggests that a seemingly simple computation can be implemented in a multisynaptic manner, allowing for greater functional robustness.

## Results

### ON pathway medulla neurons receive graded, glutamatergic input

To test if medulla neurons in the ON pathway receive glutamatergic input resembling the L1 response, we used the genetically encoded glutamate sensor iGluSnFR (*Marvin et al., 2013*; *Richter et al., 2018*). We selectively expressed iGluSnFR in the two major postsynaptic targets of L1: the medulla neurons Mi1 and Tm3. Using in vivo two-photon imaging, we measured visually

evoked responses on Mi1 and Tm3 dendrites reflecting their glutamatergic inputs in medulla layers M1 and M5. We also recorded iGluSnFR signals in the neurons' output layer, M9/10. In M1 and M5, both Mi1 and Tm3 neurons showed an increase in iGluSnFR signal in response to light OFF and a decrease in iGluSnFR signal in response to light ON, showing that rectification happens downstream of the glutamatergic input (*Figure 1C,D*). These signals were of the same polarity as intracellular calcium signals recorded within the presynaptic L1 axon terminals, and of the opposite polarity to calcium signals in the same layers of Mi1 and Tm3 (*Figure 1B–D*). In the proximal medulla (layer M9/10), Mi1 and Tm3 neurons showed weak iGluSnFR signals that increased in response to both light ON and OFF (*Figure 1C,D*). This data shows that the major postsynaptic targets of L1 receive glutamatergic input that provides information about both ON and OFF signals, which is in line with graded inputs coming from the L1 input to the ON pathway. Other glutamatergic inputs might further shape medulla neuron properties in the proximal medulla.

## ON responses are lost at PTX concentrations affecting GABA$_A$Rs and GluCls

We hypothesized that glutamatergic inhibition mediates the sign inversion between the dendritic extracellular glutamate signals and intracellular calcium or voltage signals measured in these neurons (*Figure 1*; *Behnia et al., 2014*). Glutamatergic inhibition can be mediated either by metabotropic glutamate receptors or by ionotropic glutamate-gated chloride channels (*Collins et al., 2012*; *Cully et al., 1996*; *Liu and Wilson, 2013*; *Parmentier et al., 1996*). To determine which of these receptor types mediates ON responses, we recorded in vivo calcium signals in response to visual stimuli in the two major postsynaptic partners of L1 while pharmacologically inhibiting each of the two receptor classes. When flies expressing GCaMP6f in Mi1 neurons were shown 5 s full-field flashes, Mi1 showed a transient increase in calcium signals in response to the ON step that decayed to reach a plateau response within 2 s (*Figure 2—figure supplement 1A*). Bath application of 2-Methyl-6-(phenylethenyl) pyridine hydrochloride (MPEP), a selective blocker of metabotropic glutamate receptors, to the same flies did not reduce the responses to visual light flashes in Mi1 neurons (*Figure 2—figure supplement 1A,B*). Before drug application, Tm3 responses to ON flashes showed a transient light response. Similar to Mi1, Tm3 responses were not affected by MPEP application (*Figure 2—figure supplement 1C,D*).

We next applied picrotoxin (PTX), a drug that is known to inhibit glutamate-gated chloride channels at high concentrations (*Cully et al., 1996*; *Etter et al., 1999*), but which also affects GABA$_A$Rs at much lower concentration (*Takeuchi and Takeuchi, 1969*). In vivo studies in *Drosophila* had previously used concentrations of 1–5 µM PTX to effectively block GABA-gated hyperpolarization in the olfactory system and GABAergic inhibition in *Drosophila* visual system neurons (*Fisher et al., 2015b*; *Wilson and Laurent, 2005*). In contrast, 100 µM PTX was used to block GluCls in the olfactory system (*Liu and Wilson, 2013*). Upon bath application of 100 µM PTX, visual responses were completely abolished in Mi1 neurons (*Figure 2A*). Surprisingly, when we tested the effect of low concentrations (2.5 µM) of PTX, ON responses were also lost in Mi1 (*Figure 2A*). To test this effect more precisely, we used a range of PTX concentrations and observed a loss of visual responses at concentrations ranging from 2.5 µM to 100 µM PTX in Mi1 (*Figure 2B*). When we performed the same experiments in Tm3, we again found all ON responses to be eliminated in 100 µM PTX and strongly reduced at low concentrations of PTX (*Figure 2C,D*). This effect was consistent across all medulla layers (*Figure 2A–D*, *Figure 2—figure supplement 2A–D*). Dendritic Mi1 and Tm3 regions even showed a small decrease in calcium in response to light at the highest PTX concentrations (*Figure 2—figure supplement 2A–D*).

To evaluate if the PTX effect is ON-pathway selective and to measure the compound effect on ON and OFF pathway responses, we next imaged calcium signals in the direction-selective T4 and T5 axon terminals, the stage at which many medulla neuron inputs converge. When we measured flash responses in T4/T5 cells expressing GCaMP6f, these neurons hardly showed any response to full-field flashes before toxin application, due to surround inhibition (*Figure 2E,F*; *Fisher et al., 2015a*). Upon bath application of 2.5 µM PTX, these flash responses were disinhibited, and T4/T5 neurons responded with an increase in calcium signal to both light ON and OFF (*Figure 2E,F*; *Fisher et al., 2015b*). Thus, T4/T5 neurons still show flash responses under conditions in which all responses of their predominant Mi1 and Tm3 inputs are abolished (*Figure 2A–D*). This suggests,

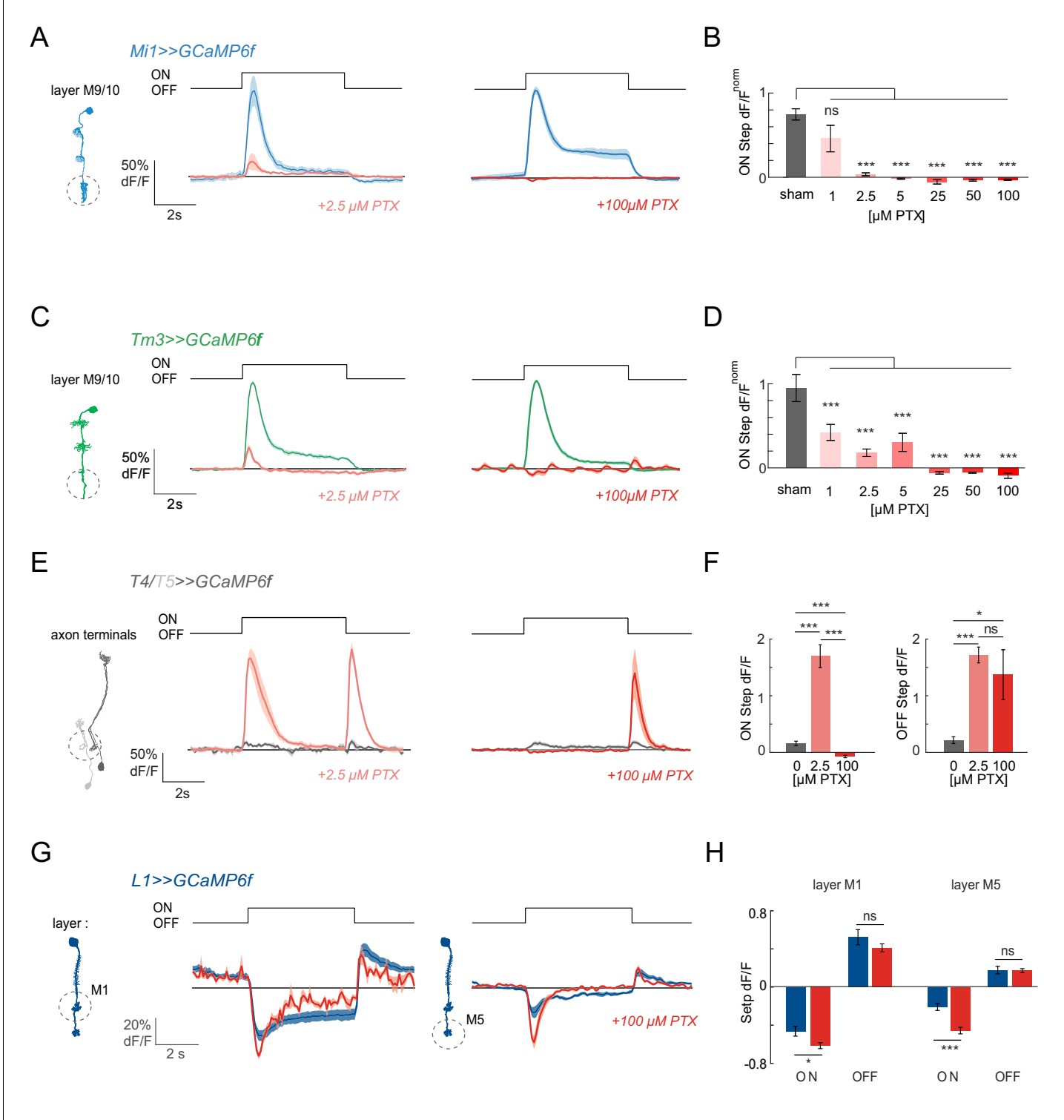

**Figure 2.** ON responses are abolished by PTX concentrations affecting GABA$_A$Rs and GluCls. (**A**) In vivo calcium signals recorded in layer M9/10 of Mi1 neurons, before (blue) and after (light red) PTX application. (**B**) Bar plot showing the quantification of the effect of PTX at various concentrations. A two-tailed Student $t$ test was performed for each concentration against the sham control. Sample sizes were as follows: sham, n = 5 (89); 1 µM PTX, n = 5 (68); 2.5 µM PTX, n = 8 (102); 5 µM PTX, n = 5 (64); 25 µM PTX, n = 4 (30); 50 µM PTX, n = 5 (87); 100 µM PTX, n = 5 (89). (**C**) In vivo calcium signals recorded in layer M9/10 of Tm3 neurons, before (green) and after (light red) PTX application. (**D**) Bar plots showing the quantification of the effect of PTX at various concentrations. A two-tailed Student $t$ test was performed for each concentration against the sham control. Sample sizes were as follows: sham, n = 7 (84); 1 µM PTX, n = 8 (108); 2.5 µM PTX, n = 9 (127); 5 µM PTX, n = 7 (74); 25 µM PTX, n = 5 (64); 50 µM PTX, n = 5 (51); 100 µM PTX, n = 6

*Figure 2 continued on next page*

*Figure 2 continued*

(92). (**E**) In vivo calcium signals in response to full-field flashes recorded in the axon terminals of T4/T5 neurons (n = 9 [229]), before (gray) and after (red) PTX application. (**F**) Bar plots showing the quantification of the results in (**E**). (**G**) In vivo calcium signals recorded in layer M1 (n = 6 [99]) and M5 (n = 6 [103]) of L1 neurons, before (dark blue) and after (red) 100 µM PTX application. (**H**) Bar plot showing the quantification of the data shown in (**G**). All traces show mean ± SEM. Sample sizes are given as number of flies (number of cells). *p<0.05, **p<0.01, ***p<0.001, tested with a one-way ANOVA and a post-hoc unpaired t-test with Bonferroni-Holm correction for multiple comparisons in (**B,D,F**), and a paired Student *t* test in (**H**).

DOI: https://doi.org/10.7554/eLife.49373.004

The following source data and figure supplements are available for figure 2:

**Source data 1.** Table 1 contains all mean ± s.e.m.
DOI: https://doi.org/10.7554/eLife.49373.011
**Figure supplement 1.** Blocking metabotropic glutamate receptors does not abolish ON responses.
DOI: https://doi.org/10.7554/eLife.49373.005
**Figure supplement 1—source data 1.** Table 1 contains all mean ± s.e.m.
DOI: https://doi.org/10.7554/eLife.49373.006
**Figure supplement 2.** ON responses are abolished by PTX concentrations affecting GABA$_A$Rs and GluCls.
DOI: https://doi.org/10.7554/eLife.49373.007
**Figure supplement 2—source data 1.** Table 1 contains all mean ± s.e.m.
DOI: https://doi.org/10.7554/eLife.49373.008
**Figure supplement 3.** The glutamatergic input onto Mi1 and Tm3 dendrites is still present upon PTX application.
DOI: https://doi.org/10.7554/eLife.49373.009
**Figure supplement 3—source data 1.** Table 1 contains all mean ± s.e.m.
DOI: https://doi.org/10.7554/eLife.49373.010

that at least under low PTX concentrations, other neurons (*Takemura et al., 2017*) and a lack of local inhibition (*Mauss et al., 2015*) can contribute to T4 responses.

After increasing the concentration of PTX to 100 µM within the same fly, all ON responses were abolished (*Figure 2E,F*), showing that T4 no longer receives any functional inputs at high PTX concentrations. In contrast, OFF responses were unaffected relative to the 2.5 µM phenotype (*Figure 2E,F*), arguing that the effect on the ON pathway is specific. This is in line with the idea that glutamate-gated chloride channels mediate ON responses in the visual system.

Importantly, the L1 input still responded to visual stimuli even at the highest PTX concentrations used (*Figure 2G,H*) and the iGluSnFR signal on the dendrites of Mi1 or Tm3 were largely unaltered, or even slightly increased at concentrations at which all Mi1 and Tm3 calcium responses were abolished (*Figure 2—figure supplement 3*), demonstrating that the glutamatergic input to the ON pathway was still intact. Taken together, our findings show that a systemic disruption of glutamate-gated chloride channels abolishes ON responses in Mi1 and Tm3, and suggest that GABA$_A$ receptors might play a role in mediating ON responses at the L1 to Mi1/Tm3 synapses in the fly visual system.

## Glutamate- and GABA-gated chloride channels are broadly expressed in the visual system

To explore the possibility that both glutamate- and GABA-gated chloride channels mediate ON selectivity, we first looked at the expression of candidate genes. The only glutamate-gated chloride channel in the fly genome is encoded by the *GluClα* gene. A GluClα protein tagged with GFP (GluClα[MI02890.GFSTF.2]) was found to be widely expressed in the visual system, including the lamina, medulla, lobula and lobula plate (*Figure 3A*). Expression was stronger in some proximal medulla layers, but the broad expression of this GFP trap did not allow expression to be assigned to specific cell types (*Figure 3A*). Two recently published cell-type-specific RNA sequencing datasets allowed us to assess candidate gene expression at cellular resolution (*Davis et al., 2018*; *Konstantinides et al., 2018*). *GluClα* mRNA was strongly expressed in all major ON pathway medulla neurons: Mi1, Tm3, Mi4, and Mi9 (*Figure 3B*, *Figure 3—figure supplement 1*). Furthermore, *GluClα* was also expressed in OFF pathway neurons (Tm1, Tm2, Tm4, and Tm9), albeit weaker in Tm9 (*Figure 3B*, *Figure 3—figure supplement 1*). The metabotropic glutamate receptor mGluR did not show expression in all ON pathway medulla neurons (*Figure 3B*, *Figure 3—figure supplement 1*), consistent with mGluR not playing a broad role in mediating ON selectivity (*Figure 2—figure supplement 1*). Of the three genes known to encode GABA$_A$Rs, *Grd* mRNA was not detectable

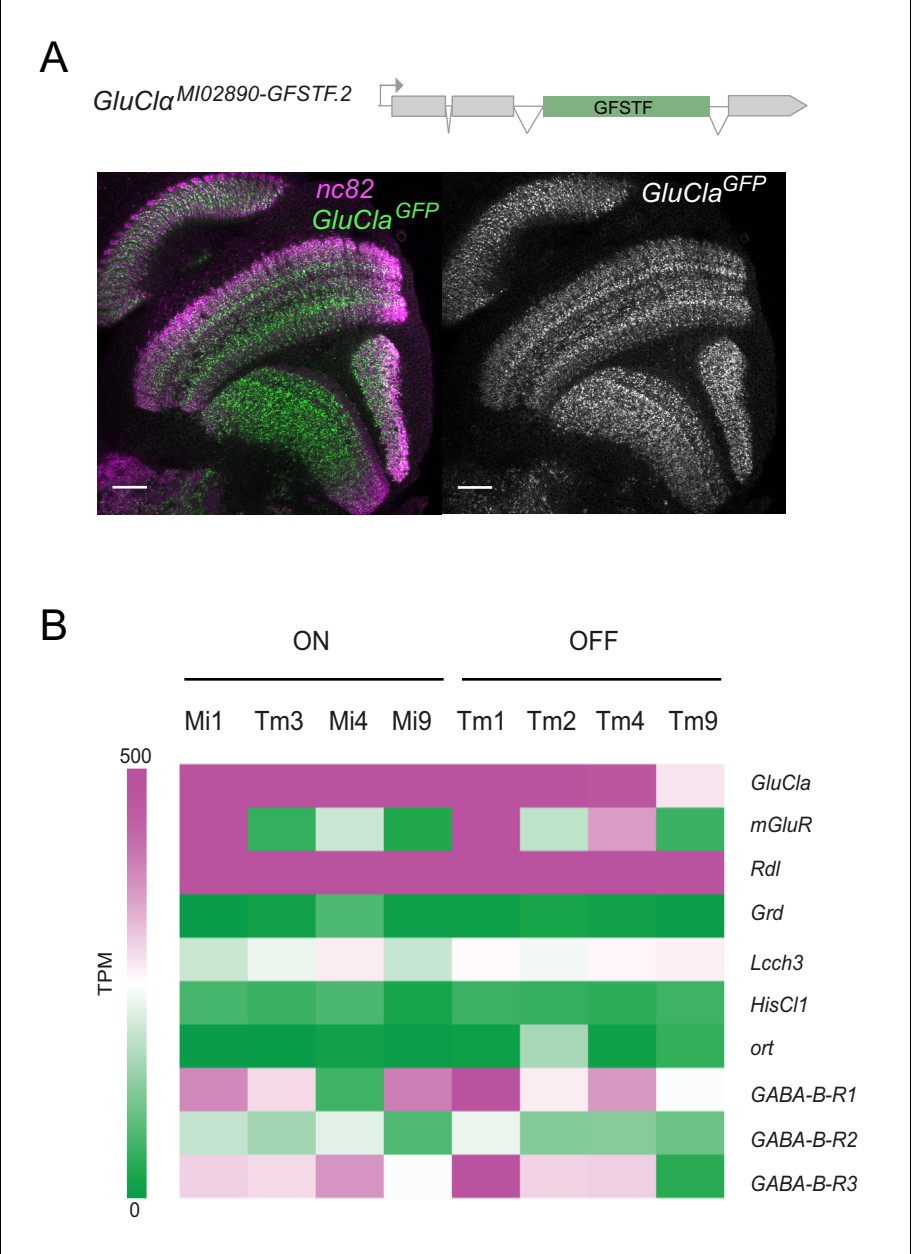

**Figure 3.** *GluClα* and *Rdl* are broadly expressed in the visual system. (**A**) Confocal cross-section of the visual system of a fly carrying a GFP exon trap within the *GluClα* locus (GluClα^MI02890.GFSTF.2). The neuropil is marked with nc82 (magenta) and endogenous GFP is in green/gray. Scale bar is 20 μm. (**B**) Expression levels shown as TPM (transcripts per kilobase million) values of inhibitory glutamate, GABA and histamine receptors. RNAseq data are from *Davis et al. (2018)* (GEO accession number: GSE 116969). Expression in the four most prominent medulla interneurons of the ON and OFF pathways are depicted.

DOI: https://doi.org/10.7554/eLife.49373.012

The following figure supplement is available for figure 3:

**Figure supplement 1.** *GluClα* and *Rdl* are broadly expressed in the visual system.

DOI: https://doi.org/10.7554/eLife.49373.013

in medulla neurons and *Lcch3* was only weakly expressed. Interestingly, the *Rdl* gene was strongly expressed in all major ON and OFF pathway medulla interneurons (*Figure 3B*). Thus, the glutamate- and GABA-gated chloride channel *GluClα* and *Rdl* are widely expressed in the visual system, including all ON pathway medulla neurons.

## A PTX-insensitive GABA$_A$R allele partially rescues ON responses

We next wanted to determine whether the PTX-induced loss of responses was due to inhibition of the glutamate receptor GluClα, the GABA receptor Rdl, or both. We therefore added molecular specificity to the pharmacological approach using alleles that are insensitive to PTX. For *Rdl*, a single point mutation has been described that leaves channel function intact but renders GABA$_A$R insensitive to PTX (*Ffrench-Constant et al., 1993*). We hypothesized that if ON responses are mediated by the Rdl receptor, the PTX-insensitive *Rdl*$^{MDRR}$ allele should rescue the effect of PTX on visual responses in ON pathway medulla neurons. To ensure that Rdl channels were exclusively composed of the PTX-insensitive Rdl$^{MDRR}$ subunit, experiments were performed in trans to an *Rdl* null mutant (*Rdl$^1$/Rdl$^{MDRR}$*), or in homozygosity (*Rdl$^{MDRR}$/Rdl$^{MDRR}$*). We tested rescue of visual responses by the *Rdl*$^{MDRR}$ mutant at 2.5 μM PTX, as this was the lowest toxin concentration that resulted in a loss of ON responses in both Mi1 and Tm3. We individually quantified the amplitude of the maximum response to the ON step, the amplitude of the plateau response, and the integrated response during the ON step (*Figure 4—figure supplement 1A*).

Control Mi1 neurons showed significantly reduced Mi1 peak responses and an eliminated sustained component upon application of PTX (2.5 μM), similar to PTX application in wild type (*Figure 4A*, *Figure 4—figure supplement 1B*). Importantly, when channels were only composed of the *Rdl*$^{MDRR}$ insensitive subunit, Mi1 responses were partially rescued and the sustained component of the response was present (*Figure 4B,C Figure 4—figure supplement 1C*). Whereas the rescue of the peak ON response was only significant in layer M1 (*Figure 4C*), the integrated response or the plateau response were also prominently rescued in other layers (*Figure 4—figure supplement 1D*). In Tm3 neurons, PTX application also significantly reduced ON responses in controls (*Figure 4D*). This response was partially rescued by the presence of the PTX insensitive *Rdl*$^{MDRR}$ allele (*Figure 4E*). This effect was again strongest in layer M1 (*Figure 4E,F*, *Figure 4—figure supplement 1E–G*). The fact that the *Rdl*$^{MDRR}$ allele does not fully rescue all ON responses in Mi1 or Tm3 suggests that PTX might also be acting on *GluClα* in this context. At the same time, our findings argue that responses in medulla neurons Mi1 and Tm3 are indeed mediated at least in part by the GABA$_A$ receptor Rdl.

All lamina neurons downstream of photoreceptors were shown to be GABA negative by immunostaining, whereas the lamina feedback neurons C2 and C3 are GABA positive (*Kolodziejczyk et al., 2008*). RNAseq data support this notion, since L1 expresses high levels of genes involved in glutamate synthesis and does not express any GABA synthesis enzymes (*Figure 4—figure supplement 2A*; *Davis et al., 2018*). Expression of the vesicular GABA transporter *dVGAT* appears high, but this gene is highly expressed in all neurons, and could be non-specific. Furthermore, although it has been shown that neurons can maintain inhibitory signaling via uptake of GABA (*Tritsch et al., 2014*), this requires expression of the plasma membrane GABA transporter Gat, which is again not expressed in L1 (*Figure 4—figure supplement 2A*). Finally, GABA immunostaining is not visible in the terminals or cell bodies of L1 neurons, but can be seen in C2/C3 neurons (*Figure 4—figure supplement 2B,C*; *Kolodziejczyk et al., 2008*). Thus, there is no evidence for L1 co-releasing GABA in addition to glutamate. This suggests that visual responses to ON stimuli in Mi1 and Tm3 do not arise solely through a monosynaptic connection with the L1 inputs as previously thought (*Figure 4Gi*), but that a GABAergic synapse involving Rdl is likely involved in circuitry upstream of Mi1 and Tm3. In summary, Rdl-dependent circuits parallel to the glutamatergic L1 to medulla neuron synapse can also mediate ON responses (*Figure 4Gii*).

## A PTX-insensitive GluClα partially rescues ON responses

We next wanted to test if and to what degree *GluClα* contributes to visual ON responses. No PTX-insensitive *GluClα* allele has been isolated in *Drosophila*, but the crystal structure of *GluClα* has been solved for the *C. elegans* homolog and the amino acid side chains interacting with the toxin have been identified (*Hibbs and Gouaux, 2011*). PTX interacts with specific residues of the M2 transmembrane alpha helix (*Figure 5A*; *Hibbs and Gouaux, 2011*). We aligned the M2 amino acid sequences of histamine, glutamate or GABA-gated chloride channels from different species (*D. melanogaster, C. elegans, M. domestica*) with known PTX sensitivities (*Cully et al., 1994*; *Ffrench-Constant et al., 1993*; *Hirata et al., 2008*; *Horoszok et al., 2001*; *Zheng et al., 2002*). This region is highly conserved among different channels and among species, with the exception of a single variable amino

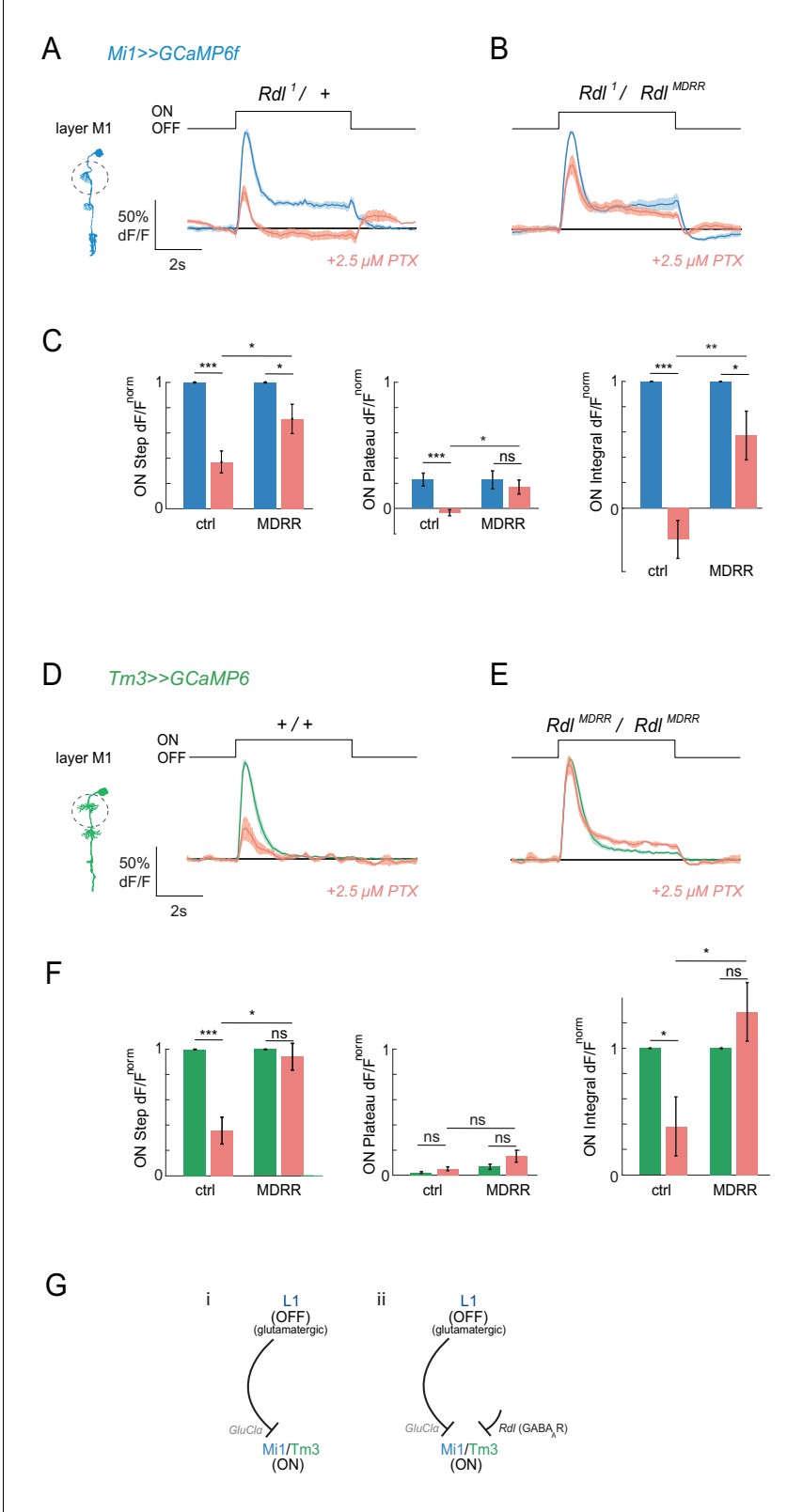

**Figure 4.** Pharmacogenetics shows that ON responses are partially mediated by the GABA$_A$R Rdl. **(A,B)** In vivo calcium signals in response to full-field flashes recorded in layer M1 of Mi1 neurons. Figure shows traces before (blue) and after (red) 2.5 μM PTX application in heterozygous $Rdl^1/+$ controls (A, n = 7 (58)), or flies only expressing the PTX-insensitive $Rdl^{MDRR}$ allele ($Rdl^1/Rdl^{MDRR}$) (B, n = 8 (77)). **(C)** Bar plots showing the quantification of the

*Figure 4 continued on next page*

*Figure 4 continued*

normalized ON step, ON plateau and ON integral of the data shown in (A,B). *p<0.05, **p<0.01, ***p<0.001, tested with an unbalanced two-way ANOVA, corrected for multiple comparisons. (D,E) In vivo calcium signals in response to full-field flashes recorded in the layer M1 of Tm3 neurons. Genotypes: ctrl = $Rdl^1$/+ and 1 = $Rdl^1$/$Rdl^{MDRR}$. Figure shows traces before (green) and after (red) PTX application +/+ controls (D, 1 = 5[34]), or flies only expressing the PTX-insensitive $Rdl^{MDRR}$ allele ($Rdl^{MDRR}$/$Rdl^{MDRR}$) (E, n = 5[46]). (F) Bar plots showing the quantification of the traces shown in (D,E). All traces show mean ± SEM. Sample sizes are given as number of flies (number of cells). *p<0.05, **p<0.01, ***p<0.001, tested with an unbalanced two-way ANOVA, corrected for multiple comparisons. (G) Schematic summarizing the results. Our results suggest that ON-selectivity does not arise solely through glutamate-gated chloride channels as initially thought (i). The GABA$_A$R Rdl is required for ON-responses in a pathway parallel to the monosynaptic L1-Mi1/Tm3 connection (ii).

DOI: https://doi.org/10.7554/eLife.49373.014

The following source data and figure supplements are available for figure 4:

**Source data 1.** Table 1 contains all mean ± s.e.m.

DOI: https://doi.org/10.7554/eLife.49373.018

**Figure supplement 1.** Pharmacogenetics shows that ON responses are partially mediated by the GABA$_A$R Rdl.

DOI: https://doi.org/10.7554/eLife.49373.015

**Figure supplement 1—source data 1.** Table 1 contains all mean ± s.e.m.

DOI: https://doi.org/10.7554/eLife.49373.016

**Figure supplement 2.** L1 neurons are not GABAergic.

DOI: https://doi.org/10.7554/eLife.49373.017

acid, corresponding to amino acid S278 in *D. melanogaster* GluClα (red, *Figure 5B*). The identity of this single amino acid correlates strongly with the PTX sensitivity of the channel (*Figure 5B*). Mutations in this amino acid have been shown to change the PTX sensitivity of the channel. For example, the A > S substitution in the *D. melanogaster* GABA$_A$R allele $Rdl^{MDRR}$ exhibits reduced sensitivity to PTX (*Figure 4*, *Ffrench-Constant et al., 1993*; *Fisher et al., 2015a*).

This prompted us to generate a potentially PTX-insensitive version of *GluClα* by introducing a point mutation leading to an S278T exchange. We first characterized mutant *GluClα*$^{S278T}$ heterologously in *Xenopus* oocytes. Two-electrode voltage-clamp recordings of wild type *GluClα*-expressing oocytes revealed fast activating and rapidly inactivating glutamate-induced currents, similar to inhibitory glutamate currents recorded in vivo in honeybees (*Barbara et al., 2005*). This current was sensitive to PTX (*Figure 5C,E*). Expression of the *GluClα*$^{S278T}$ mutant led to glutamate-induced currents that were less inactivating compared to wild-type *GluClα* controls. Importantly, the glutamate-induced currents in the *GluClα*$^{S278T}$ mutant were insensitive to PTX (*Figure 5D,E*).

We next generated *GluClα*$^{S278T}$ mutant flies, targeting the endogenous *GluClα* gene locus using CRISPR/Cas9-based genome editing (see Materials and methods for details). In the absence of toxin, *GluClα*$^{S278T}$ flies responded to visual stimuli with the typical peak and plateau response, arguing that the altered kinetics of the *GluClα*$^{S278T}$ mutant observed in oocytes was not a problem under these stimulus conditions (*Figure 6A,B*). We then tested if the *GluClα*$^{S278T}$ allele could rescue PTX-induced phenotypes in vivo in the visual system. Because the PTX-insensitive $Rdl^{MDRR}$ allele rescued visual responses only partially at 2.5 µM, we first tested if *GluClα* could also account for a loss of responses at such low PTX concentrations previously thought to only block GABA$_A$Rs. Upon application of low concentrations of PTX (2.5 µM), calcium responses were lost in Mi1 neurons of heterozygous *GluClα* controls carrying a deficiency (Df) uncovering the *GluClα* locus (*Figure 6A*). In flies only expressing the *GluClα*$^{S278T}$ and no wild type protein, the ON responses were partially rescued. The rescue was specifically prominent for the step response, which was significantly rescued by *GluClα*$^{S278T}$ in all medulla layers (*Figure 6B,C*, *Figure 6—figure supplement 1A–C*). This shows that GluClα in vivo is sensitive to lower concentrations of PTX than previously thought, arguing that there is no specific concentrations to only block Rdl, and highlighting the usefulness of these PTX-insensitive alleles for molecular specificity. Furthermore, the rescue by *GluClα*$^{S278T}$ demonstrates that GluClα mediates ON responses in Mi1 in the fly visual system.

When testing if *GluClα*$^{S278T}$ could also rescue Tm3 responses in the presence of the toxin, this batch of 2.5 µM PTX gave a comparably mild phenotype in heterozygous controls (*Figure 6D*). The integrated ON response was still significantly rescued in a *GluClα*$^{S278T}$ background in layer M1

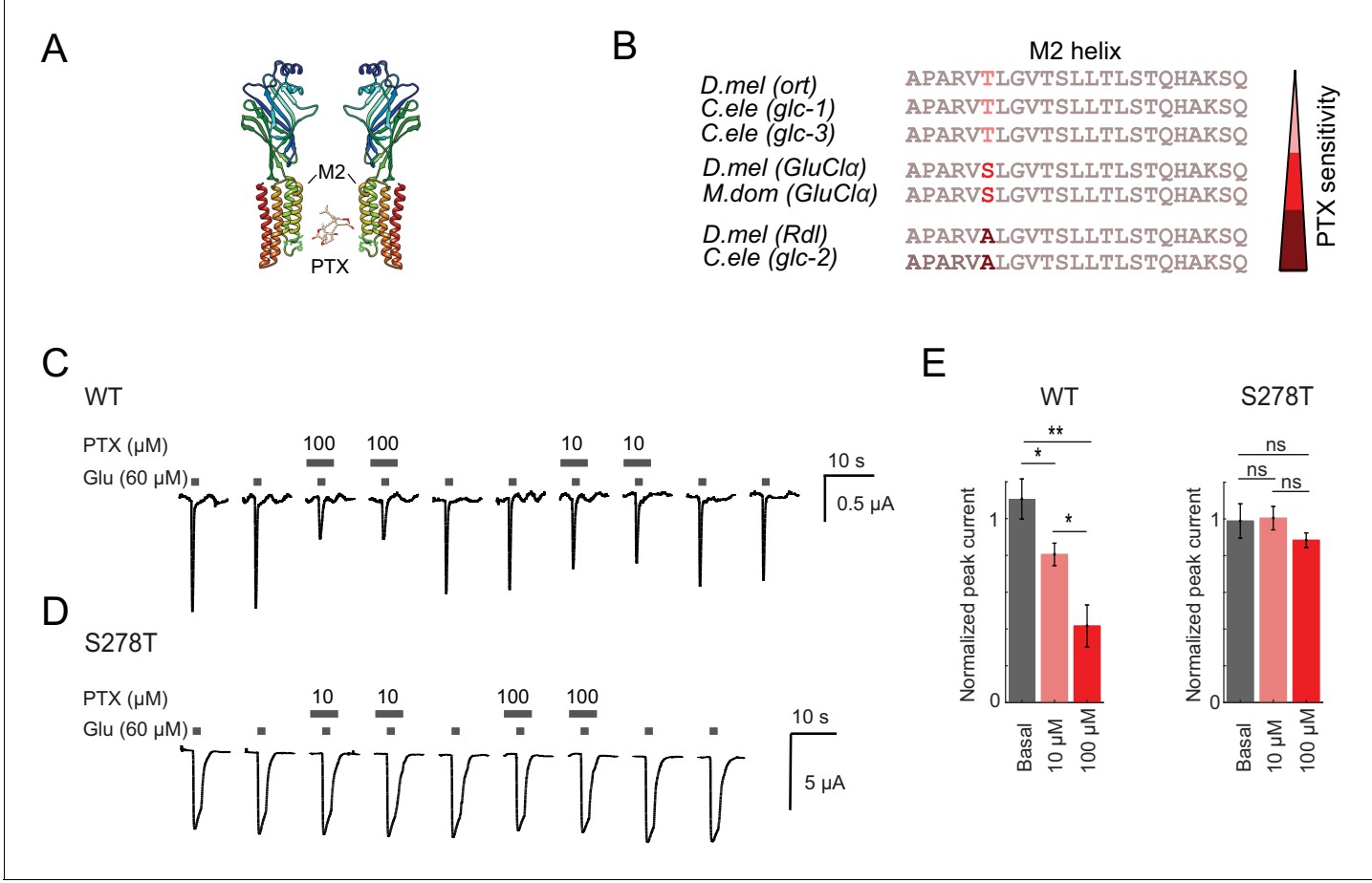

**Figure 5.** A *GluClα* allele insensitive to picrotoxin. (**A**) 3D protein structure of *Drosophila melanogaster* GluClα binding PTX, obtained by homology modeling with its *C. elegans* homolog GluClα using the Protein Homology/analogY Recognition Engine Phyre 2. (*Kelley et al., 2015*). The PTX structure was obtained from DrugBank, identification number DB00466. Structures were edited using Chimera 1.13, (*Pettersen et al., 2004*). (**B**) Alignment of the M2 helix of different ligand-gated chloride channel. The histamine-gated chloride channel *ort* (*Zheng et al., 2002*), the glutamate-gated chloride channels *glc-1, glc-2, glc-3, GluClα*, (*Cully et al., 1996*; *Cully et al., 1994*; *Horoszok et al., 2001*), and the GABA$_A$R *Rdl*. PTX sensitivities are indicated as shades of red. *D.mel* = *Drosophila melanogaster*, *C.ele* = *Caenorhabditis elegans*, *M.dom* = *Musca domestica*. (**C,D**) Two-electrode voltage-clamp recordings at a holding potential of −70 mV from *X. laevis* oocytes expressing wild type (**C**) or S278T (**D**) *GluClα*. Currents were evoked by glutamate wash in (lower bars) in the absence or presence of 10 μM or 100 μM picrotoxin (upper bars). (**E**) Mean peak-current amplitudes of the glutamate-evoked response in the presence (red) and absence (gray) of picrotoxin, normalized to the peak-current amplitude evoked by glutamate after picrotoxin wash out. Bars show mean ± SEM. *p<0.05, **p<0.01, tested with a one-way ANOVA and a post-hoc unpaired t-test with Bonferroni-Holm correction for multiple comparisons. Sample sizes: WT n = 6 and S278T n = 4 for 10 μM, and WT n = 6 and S278T n = 6 for 100 μM PTX.

DOI: https://doi.org/10.7554/eLife.49373.019

The following source data is available for figure 5:

**Source data 1.** Table 1 contains all mean ± s.e.m.
DOI: https://doi.org/10.7554/eLife.49373.020

(*Figure 6E,F*). In other medulla layers, 2.5 μM PTX more prominently blocked the peak Tm3 ON response in controls, but not in a *GluClα*[S278T]-insensitive background (*Figure 6—figure supplement 1D–F*). This shows that *GluClα*[S278T] also partially mediates ON responses in Tm3.

At high concentrations of PTX (100 μM), the PTX-insensitive *GluClα*[S278T] allele did not rescue ON responses in either Mi1 or Tm3 (*Figure 6—figure supplement 1G–L*). This could suggest that GluClα[S278T] does not confer PTX sensitivity at such high PTX concentrations in vivo. Alternatively, if GluClα[S278T] was fully insensitive, this would further argue that PTX blocks other channels that are required for ON responses, and would thus underline the importance of Rdl. While we cannot fully distinguish between these two possibilities, we argued that if the loss of Mi1 and Tm3 responses

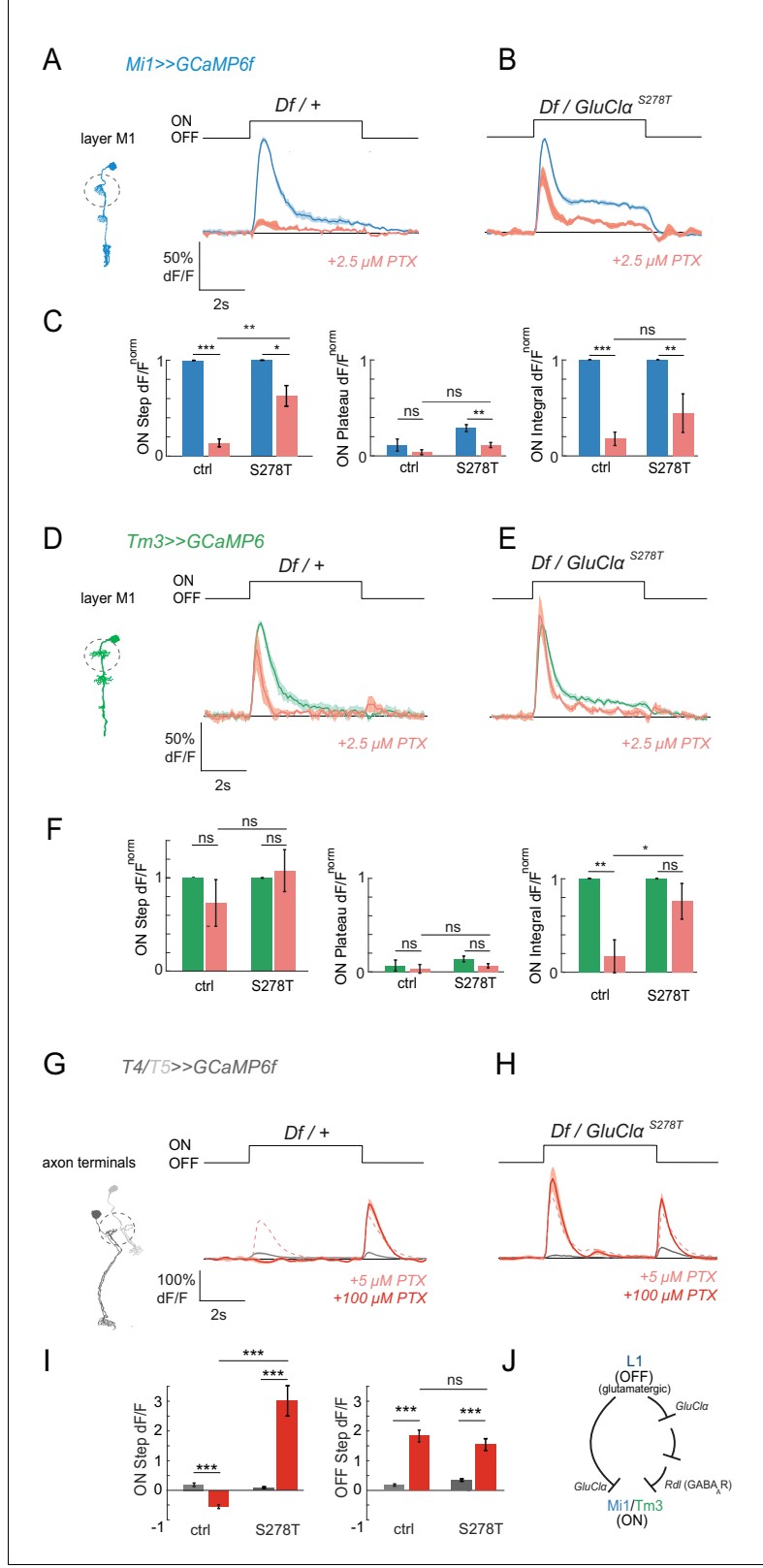

**Figure 6.** Pharmacogenetics shows that ON responses are mediated by *GluClα*. (**A,B**) In vivo calcium signals in response to full-field flashes recorded in layer M1 of Mi1 neurons. Figure shows traces before (blue) and after (red) 2.5 µM PTX application in heterozygous *GluClα*$^{Df}$/+ deficient controls (A, n = 5 [63]), as well as flies only expressing the PTX-insensitive *GluClα*$^{S278T}$ allele (*GluClα*$^{S278T}$/*GluClα*$^{Df}$).(B, n = 5 [47]). (**C**) Bar plots showing the

*Figure 6 continued on next page*

*Figure 6 continued*

quantification of the data from (**A,B**). *p<0.05, **p<0.01, ***p<0.001, tested with an unbalanced two-way ANOVA, corrected for multiple comparisons. (**D,E**) In vivo calcium signals in response to full-field flashes recorded in the layer M1 of Tm3 neurons. Figure shows traces before (green) and after (red) PTX application in heterozygous *GluClα*$^{Df}$/+ deficient controls (D, n = 5 [30]), as well as flies only expressing the PTX-insensitive *GluClα*$^{S278T}$ allele (*GluClα*$^{S278T}$/*GluClα*$^{Df}$) (E, n = 5 [40]). (**F**) Bar plots showing the quantification of the data shown in (**D,E**). *p<0.05, **p<0.01, ***p<0.001, tested with an unbalanced two-way ANOVA, corrected for multiple comparisons. (**G,H**) In vivo calcium signals in response to full-field flashes recorded in T4/T5 axon terminals. Figure shows traces before (gray) and after (red) 100 µM PTX application in heterozygous *GluClα*$^{Df}$/+ deficient controls (G, n = 10[440]), as well as flies only expressing the PTX-insensitive *GluClα*$^{S278T}$ allele (*GluClα*$^{S278T}$/*GluClα*$^{Df}$) (H, n = 5[192]). The pink dotted line shows responses after the application of 5 µM PTX. (**I**) Bar plots showing the quantification of the data shown in (**G,H**). *p<0.05, **p<0.01, ***p<0.001. Statistics was done using an unbalanced two-way ANOVA, corrected for multiple comparisons. All traces show mean ± SEM. Sample sizes are given as number of flies (number of cells). (**J**) Schematic summarizing the results. Our results provide support for a combinatorial role of glutamatergic and GABAergic inhibition in mediating ON responses. Since GluClα is likely to be the receptor on all neurons postsynaptic to L1, Rdl could function downstream of GluClα.

DOI: https://doi.org/10.7554/eLife.49373.021

The following source data and figure supplements are available for figure 6:

**Source data 1.** Table 1 contains all mean ± s.e.m.
DOI: https://doi.org/10.7554/eLife.49373.024
**Figure supplement 1.** Pharmacogenetics shows that ON responses are mediated by *GluClα*.
DOI: https://doi.org/10.7554/eLife.49373.022
**Figure supplement 1—source data 1.** Table 1 contains all mean ± s.e.m.
DOI: https://doi.org/10.7554/eLife.49373.023

were due to the role of Rdl, *GluClα*$^{S278T}$ might still rescue the 100 µM PTX phenotype in T4/T5. As we showed above, unlike Mi1 and Tm3 responses, ON responses in T4/T5 were specifically blocked by high but not low concentrations of PTX (*Figure 2E,F*). To test if *GluClα*$^{S278T}$ can rescue T4/T5 responses at high concentrations, we recorded calcium signals in T4/T5 cells in a *GluClα*$^{S278T}$ background. Indeed, ON responses to full-field light flashes in 100 µM PTX were rescued (*Figure 6G–I*). Although this experiment does not tell us which cell types this rescue is coming from, this data shows that GluClα$^{S278T}$ can be effective to rescue responses in some cell types at high PTX concentrations in vivo. Thus, the use of the toxin-insensitive *GluClα*$^{S278T}$ mutant demonstrates that *GluClα* also mediates ON responses in vivo in the fly visual system. Together, our data provides support for a combinatorial role of glutamatergic and GABAergic inhibition in mediating ON responses. Because of the glutamatergic L1 input, GluClα is likely to be the receptor on all neurons postsynaptic to L1 (*Figure 6J*). Rdl could function downstream of GluClα. This suggests that a pathway parallel to a monosynaptic glutamatergic circuit can also mediate ON responses in Mi1 and Tm3 (*Figure 6J*).

## The sign inversion in the ON pathway is a multisynaptic computation that depends on GluClα

Our findings lead to a model in which *GluClα* mediates responses to glutamatergic inputs in neurons downstream of L1 and in which a GABAergic pathway additionally drives responses in the ON pathway medulla neurons Mi1 and Tm3 (*Figure 6J*). Given that there is no evidence for L1 being GABAergic, Mi1 and Tm3 responses might not depend solely on monosynaptic L1 input. If this hypothesis is correct, *GluClα* should not exclusively function in a cell-autonomous manner in neurons downstream of L1, suggesting that Mi1 and Tm3 might still be able to respond to ON signals when *GluClα* function is only disrupted within the respective cell type. However, since pharmacological perturbations always targeted the entire visual system, more specific targeting would be required to address this possibility.

To test the above hypothesis, we generated a *GluClα* loss-of-function specifically in either Mi1 or Tm3. We inserted a FlpStop exon (*Fisher et al., 2017*) in the non-disrupting orientation (*GluClα*$^{FlpStop.ND}$), in which splicing occurs normally unless the FlpStop exon is inverted by Flp recombinase expression (*Fisher et al., 2017*). Upon pan-neuronal inversion of the FlpStop cassette into the non-disrupting orientation (*GluClα*$^{FlpStop.D}$) quantification of *GluClα* expression levels using

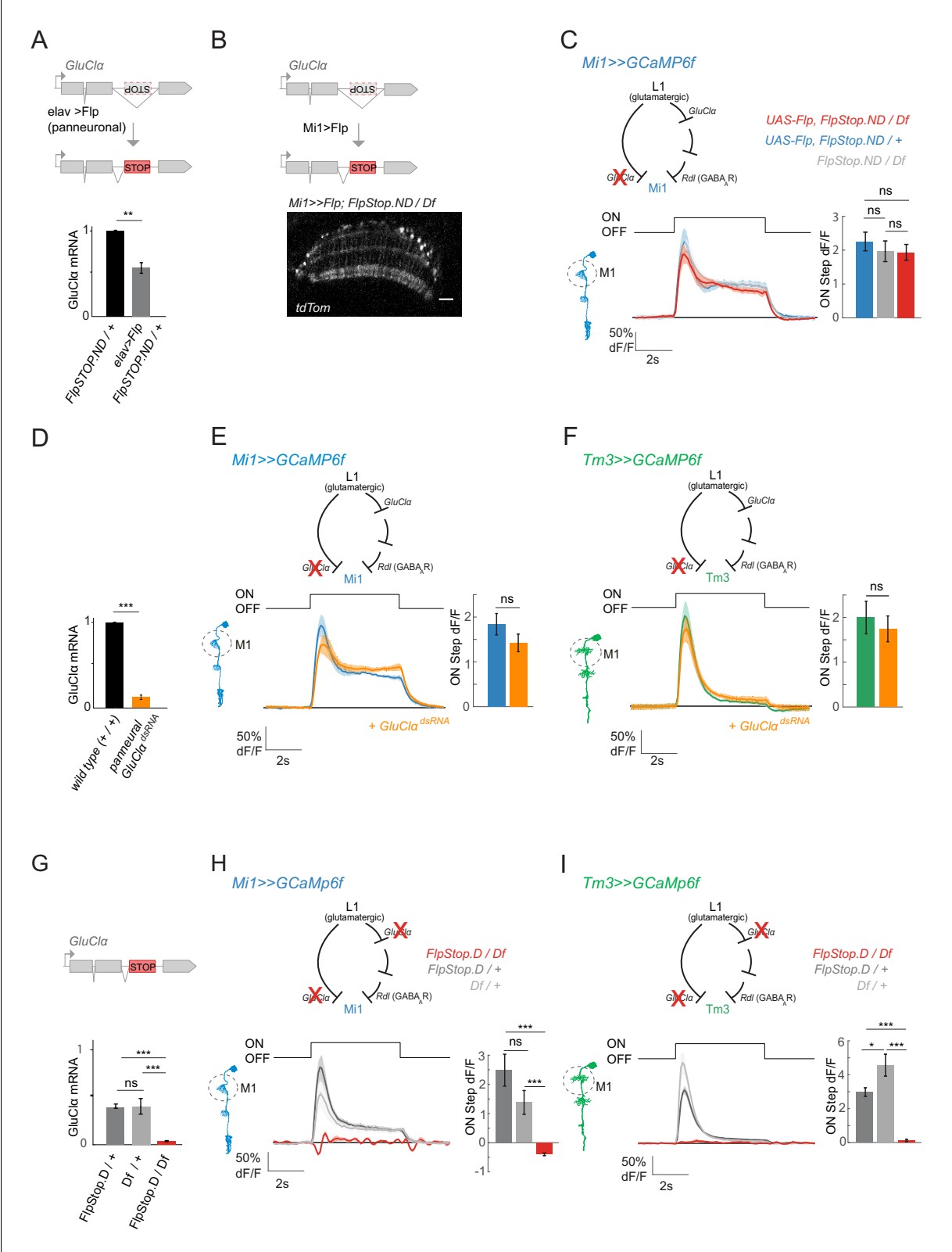

**Figure 7.** ON selectivity is a multisynaptic computation mediated by *GluClα*. (**A**) Schematic illustrating inversion of the *GluClα* FlpStop exon in all neurons by panneuronal expression of Flp recombinase using elav-Gal4. qRT-PCR results show *GluClα* mRNA levels relative to the GAPDH housekeeping gene, normalized to controls. (**B**) The FlpStop exon in the non-disrupting (ND) orientation was inverted specifically in Mi1 neurons by cell-type-specific expression of Flp recombinase. This is visualized via expression of tdTom signal, and is specific to and broad in Mi1. (**C**) In vivo

*Figure 7 continued on next page*

*Figure 7 continued*

calcium signals recorded in a cell-type-specific Mi1 *GluClα* loss-of-function. Calcium signals were recorded in layer M1 of *GluClα* mutant Mi1 neurons (n = 5 (136)) (red), and heterozygous controls (n = 5[166], n = 5[139]) (blue, gray). The bar plot shows quantification of the ON step. (D) qRT PCR results quantifying pan neuronal knockdown of *UAS-GluClα*$^{RNAi}$ using elav-Gal4, normalized to control. (E,F) In vivo calcium signals upon cell-type-specific *GluClα* knockdown. Calcium signals in response to full-field flashes were recorded in layer M1 of (E) Mi1 control (n = 8[229]) or *Mi1 >>GluClα*$^{RNAi}$ (n = 6 [215]) or of (F) Tm3 control (n = 8[150]) and *Tm3 >>GluClα*$^{RNAi}$ (n = 6[146]). Bar plot show quantification. (G) qRT PCR results quantifying *GluClα* mRNA levels in heterozygous *GluClα*$^{Flp.Stop.D/+}$ and *GluClα*$^{Df/+}$, as well as the homozygous mutant *GluClα*$^{Flp.Stop.D/Df}$. (H,I) In vivo calcium signals recorded in a full *GluClα*$^{FlpStop.D}$ mutant background (red). Heterozygous *GluClα*$^{FlpStop.D}$ controls are in blue (Mi1, n = 5 (117)) or green (Tm3, n = 8 (275)), heterozygous Df controls are in gray (Mi1, n = 5[142]; Tm3, n = 5[198]) and the experimental condition is in red (Mi1, n = 9 [134]; Tm3, n = 5[96]). Bar plots show quantification of the ON step response. All traces show mean ± SEM. All sample sizes are shown as number of flies (number of cells). Statistics was done using an unpaired Student *t* test for comparison in (A,D,E,F), and a one-way ANOVA and a post-hoc unpaired t-test with Bonferroni-Holm correction for multiple comparisons in (C,G,H,I). *p<0.05, **p<0.01, ***p<0.001.
DOI: https://doi.org/10.7554/eLife.49373.025

The following source data is available for figure 7:

**Source data 1.** Table 1 contains all mean ± s.e.m.
DOI: https://doi.org/10.7554/eLife.49373.026

qRT-PCR showed that transcription was disrupted by half when the FlpStop exon was inverted in a heterozygous background, arguing for a full loss of function in *GluClα*$^{FlpStop.D}$ (***Figure 7A***).

To selectively disrupt *GluClα* function in Mi1 neurons, we expressed Flp recombinase in Mi1 (***Figure 7B***). Expression of tdTomato, a marker for the FlpStop inversion event, further confirmed efficient inversion of the cassette (***Figure 7B***). When we recorded calcium signals in Mi1 in this cell-type-specific FlpStop background, visual responses to ON flashes were still present in Mi1 neurons and did not differ from controls (***Figure 7C***). To corroborate these findings, we next used cell-type-specific RNAi. Pan-neuronal expression of *GluClα*$^{dsRNA}$ reduced *GluClα* mRNA to 16.4 ± 7.4% of controls (***Figure 7D***). When knocking down *GluClα* in either Mi1 or Tm3, ON responses were again not abolished and did not differ significantly from controls (***Figure 7E,F***). These results demonstrate that ON selectivity is not mediated monosynaptically, but that pathways parallel to the L1-Mi1 or L1-Tm3 connection might be sufficient to mediate ON responses.

Our results argue that ON responses are encoded in a multi-synaptic fashion. Because L1 is the major input to the ON pathway and is glutamatergic (***Figure 4—figure supplement 2A***; *Takemura et al., 2011*), all ON responses might still depend on *GluClα* at the first synapse postsynaptic to L1. This leads to the hypothesis that Mi1 or Tm3 responses would be lost in a full *GluClα* mutant background. To test this, we used a FlpStop allele inserted in the disrupting orientation. In this background, expression should be fully disrupted in all cells normally expressing *GluClα*. Quantification of expression levels of *GluClα* using qRT-PCR showed that transcription was fully disrupted in *GluClα*$^{FlpStop.D}$ (3.7 ± 0.6% mRNA compared to wild type, ***Figure 7G***). Furthermore, in heterozygous animals, *GluClα*$^{FlpStop.D}$ transcripts were reduced roughly by half (40.1 ± 2.2%) and to the same amount as in a *GluClα* deficiency (40.2 ± 8.2%) lacking the entire gene locus (***Figure 7G***). These findings confirm that *GluClα*$^{FlpStop.D}$ is a null allele.

*GluClα*$^{FlpStop.D}$ mutant flies eclosed but showed locomotor deficits. The viability of the *GluClα* mutant allowed us to conduct calcium imaging experiments in this null mutant background. Whereas Mi1 and Tm3 neurons in heterozygous *GluClα* mutant or deficient flies responded normally to light flashes, Mi1 and Tm3 responses were both dramatically affected in full *GluClα* mutants. No increase in calcium signal was detectable in *GluClα* null mutants, and light responses were largely absent in all layers (***Figure 7H,I***). Instead, Mi1 even showed a small and transient decrease in calcium signal in response to light ON, which could potentially be attributed to the additional presence of excitatory glutamate receptors or reveal inputs from the OFF pathway (***Figure 7H***). Thus, normal ON responses are lost whenever *GluClα* function is disrupted in the entire visual system, but not when it is disrupted in a cell-type-specific manner. These data are further consistent with the results of pharmacological experiments in wild type and PTX-insensitive alleles. Taken together, these results demonstrate that ON selectivity is a multisynaptic computation that is robust to perturbations at individual synapses.

## GluClα mediates all ON but not OFF responses in direction-selective cells

To generalize the role of *GluClα* function for ON responses, we next asked how the output of the system is affected in a full *GluClα* loss-of-function by recording T4/T5 responses in the *GluClα*^FlpStop.D^ mutant. When imaging flies expressing GCaMP6f in both T4 and T5, individual cell type responses can be separated by showing individual moving ON and OFF edges that activate T4 and T5, respectively (*Figure 8A*; *Fisher et al., 2015b*; *Maisak et al., 2013*). T4/T5 neurons project to one of the four layers of the lobula plate, and the four layers show distinct directional tuning. In heterozygous controls, T4/T5 neurons responded to both moving ON and OFF edges and the four layers responded preferentially to front to back (layer A), back to front (layer B), upward (layer C), and downward (layer D) motion (*Figure 8A*). Responses to ON edges were completely abolished for motion in all directions in the *GluClα* null mutant, showing that all T4 inputs depend on *GluClα* (*Figure 8B,C*). In contrast, T4/T5 neurons still responded to OFF edges moving in different directions (*Figure 8A–D*). Both response amplitude and direction selectivity of the OFF response were unaffected in *GluClα* mutants (*Figure 8D*).

When recording responses to light flashes, T4/T5 axon terminals also no longer showed an increase in calcium in response to the ON step. Interestingly, the calcium signal even decreased, indicating inhibition (*Figure 8E,G*). Inhibition was previously shown to be an important part of motion computation (*Fisher et al., 2015a*; *Gruntman et al., 2018*; *Haag et al., 2016*; *Leong et al., 2016*; *Salazar-Gatzimas et al., 2016*). This could argue that, in the absence of all ON inputs, feed-forward inhibition onto T4 is revealed. Alternatively, T5 neurons might have lost rectification and respond to ON with a decrease in calcium. To distinguish between these two possibilities, we recorded flash responses in layer M10 of the medulla. Here, T4 and T5 projections do not overlap and calcium signals will stem exclusively from T4 dendrites. We found that T4 dendrites also show a negative calcium signal in response to ON (*Figure 8F,H*), revealing that this inhibition is present in T4 neurons but masked in the presence of *GluClα*. Furthermore, there was an increase in calcium signal during OFF in T4 dendrites (*Figure 8F*), suggesting a loss of *GluClα*-dependent inhibition that is normally active during OFF. Taken together, our results show that *GluClα* function is critical for ON selectivity in the fly visual system.

## Discussion

In this study, we have identified the mechanisms underlying splitting of the ON and OFF pathways in the *Drosophila* visual system. As expected from the major input to the ON pathway being glutamatergic, broad *GluClα* function is required for all ON responses in medulla neurons or downstream direction-selective cells. However, individual cell types downstream of the glutamatergic L1 input are resilient to a cell-type-specific loss of *GluClα*, demonstrating that ON selectivity is computed in a distributed manner. We further show that both the glutamate-gated chloride channel *GluClα* and the GABA-gated chloride channel *Rdl* are widely expressed in the visual system and together mediate ON responses. Thus, ON selectivity is a multisynaptic computation that is established across distributed circuits.

### ON selectivity is mediated by both glutamatergic and GABAergic inhibition

Our work shows that visual responses in the first ON-selective neuron of the *Drosophila* visual system uses a combination of *GluClα* and *Rdl* receptors. This reveals a new biophysical mechanism through which ON and OFF pathway dichotomy can be established. While pharmacology can be used to deduce the function of specific molecular mechanisms, these approaches are often not specific to one protein. GluCls and GABARs belong to the same receptor family of ligand-gated chloride channels and have closely related structure and phylogeny (*Betz, 1990*; *Lynagh et al., 2015*). All known noncompetitive antagonists like Picrotoxin, γ-HCH, dieldrin, EBOB and fibronil target both receptor types although the actions are weaker in GluCls compared to GABARs (*Eguchi et al., 2006*). Along these lines, PTX was thought to affect GABA_A receptor at low concentrations, and additionally affect GluCls at high concentrations in vitro and in vivo (*Liu and Wilson, 2013*; *McCavera et al., 2009*; *Takeuchi and Takeuchi, 1969*; *Wilson and Laurent, 2005*). Here, the use of PTX-insensitive alleles

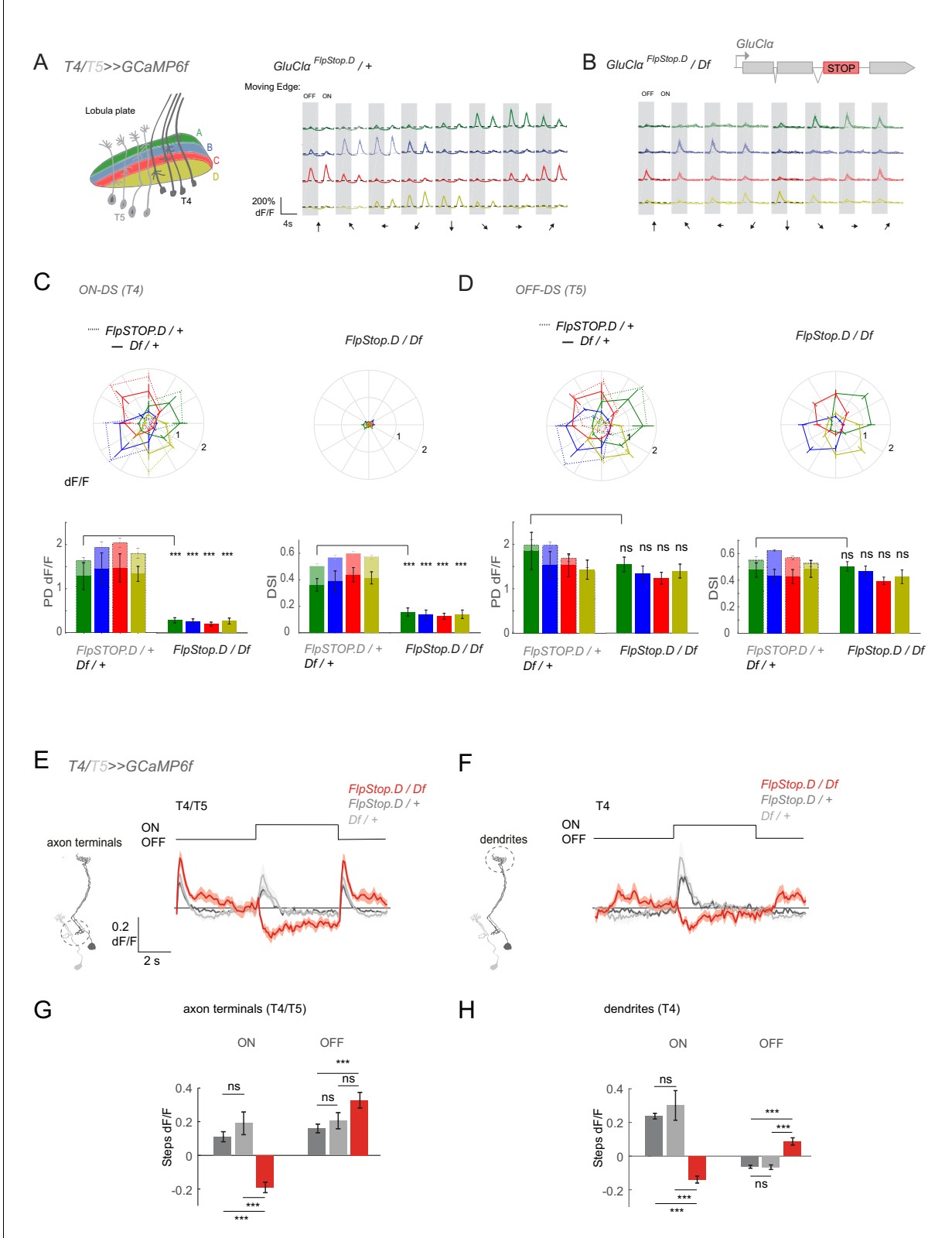

**Figure 8.** *GluClα* mediates ON but not OFF direction-selective responses. (A–D) In vivo calcium signals recorded in direction-selective T4/T5 neurons. (A,B) Schematic of T4 and T5 axon terminal innervating the four anatomical layers in the lobula plate (left). Visually evoked calcium signals recorded from all four layers in response to OFF and ON moving edges in eight different directions in heterozygous controls (A, n = 8 [62/60/59/50]) and a full *GluClα* mutant background, *GluClα^FlpStop.D^*/Df (B, n = 9 [69/73/70/64]). (C,D) Polar plots showing the calcium signals in response to moving ON (C) or

*Figure 8 continued on next page*

*Figure 8 continued*

OFF (**D**) edges. Genotypes are as indicated. Bar plots show the quantification of the preferred direction (PD) response and the direction selectivity index (DSI) for each layer. Sample size shows number of flies (number of cells in layers A/B/C/D) (**E,F**) Calcium signals in response to full-field flashes recorded in T4/T5 axon terminals in the lobula plate (**E**) or T4 dendrites in the medulla (**F**). (**G,H**) Bar plots show the quantification of the ON or OFF response in T4/T5 axon terminals (**G**) or T4 dendrites (**H**). Heterozygous *GluClα*^FlpStop.D^ (n = 8 (225/40)) and *GluClα*^Df^ (n = 8 [141/36]) controls are in gray and the *GluClα*^FlpStop.D^/Df mutant (n = 5 [198/22]) is in red in (**E–H**). All traces show mean ± SEM. Sample size are given as number of number of flies (number of ROIs). Statistical comparisons were done using an unpaired Student *t* test in (**C,D**), and a one-way ANOVA and a post-hoc unpaired t-test with Bonferroni-Holm correction for multiple comparisons in (**G,H**), ***p<0.001.
DOI: https://doi.org/10.7554/eLife.49373.027

The following source data is available for figure 8:

**Source data 1.** Table 1 contains all mean ± s.e.m.
DOI: https://doi.org/10.7554/eLife.49373.028

for glutamate and GABA-gated chloride channels allowed us to deduce that, in vivo, *GluClα* is already blocked by PTX at lower concentrations than previously thought, and that both *GluClα* and *Rdl* play critical roles for ON responses in the *Drosophila* visual system. These pharmacogenetic experiments using toxin-insensitive alleles prove to be a powerful tool to unambiguously assign specific effects to individual channels.

One benefit of the use of two inhibitory transmitter systems might be the distribution of sensory coding across parallel synapses. GluClα and Rdl also appear to have very different channel dynamics (*Cully et al., 1996*; *Ffrench-Constant et al., 1993*). Interestingly, PTX-insensitive *GluClα* and *Rdl* alleles predominantly rescue different aspects of the visual responses. Whereas *GluClα*^S278T^ predominantly rescued the peak response in all medulla layers, *Rdl*^MDRR^ mainly rescued the plateau response. This is consistent with our results and with previous oocyte recordings revealing that GluClα is fast desensitizing (*Figure 5*; *Cully et al., 1996*). It is also consistent with in vivo recordings of inhibitory glutamate currents in the honeybee (*Barbara et al., 2005*). In contrast, GABA receptors stay open throughout the period in which the transmitter is present (*Ffrench-Constant et al., 1993*). Thus, the use of different inhibitory receptors might allow different aspects of a temporally structured stimulus to be encoded. This is consistent with the finding that two different types of inhibition are also in place in the vertebrate retina. There, GABAergic and glycinergic inhibition diversify the response properties of bipolar cells through a direct influence on temporal and spatial features (*Franke et al., 2017*).

While both receptors appear to be broadly expressed in many cell types of the visual system, they could be co-expressed with different transporters and channels, and interact with different molecular partners, further diversifying their role. Another common strategy to generate functional diversity is the bringing together of different receptor subunits with certain homology. Both mammalian GlyR and GABA_A receptors can function as hetero-oligomers made up of different subunits and thus generating functional diversity (*Betz, 1990*). There are at least three different GluCl subtypes in *C. elegans* that can be combined (*Cully et al., 1994*; *Horoszok et al., 2001*). In *Drosophila*, only one gene coding for a glutamate-gated chloride channel has been identified. Although alternative splicing and post-transcriptional modifications could alter channel function, all known isoforms are identical in their functional domains. However, heteropentameric channels composed of mixed Rdl and GluClα subunits have been suggested biochemically (*Ludmerer et al., 2002*). Such a potential presence of hybrid channels might also explain the higher in vivo sensitivity of GluClα to PTX in some cell types (*Figure 6*, S7). Finally, two distinct inhibitory transmitter systems might be suitable for individual changes during evolution, allowing for adaptation to specific contextual constraints.

## ON selectivity is a multisynaptic computation

Our experiments revealed that *GluClα* is not exclusively required in a cell-autonomous manner for ON responses, since loss of *GluClα* function in Mi1 or Tm3 individually does not lead to a loss of ON responses. It is unlikely that this is due to an incomplete loss of function, since independent genetic tools (FlpStop and RNAi) that both disrupted *GluClα* expression substantially at the mRNA level (*Figure 7*) gave the same result. Furthermore, the same FlpStop allele effectively abolished all ON responses when *GluClα* function was disrupted within its entire expression pattern. Additionally, a PTX-resistant Rdl channel can mediate ON responses in a PTX background, although L1 is not

GABAergic. Together, these results suggest that ON selectivity is not a monosynaptic computation, but that parallel functional pathways can even compensate for the loss of the major synaptic connection that links L1 directly to Mi1 or Tm3. Thus, the emergence of ON selectivity is more distributed than suggested by minimal core circuit motifs. One synaptic layer further downstream, optogenetic activation of Mi1 and Tm3 most strongly contributes to T4/T5 responses (*Strother et al., 2017*). However, our data further show that T4/T5 neurons still respond to ON stimuli when both Mi1 and Tm3 responses are completely blocked by PTX, arguing that other neurons also significantly contribute to T4/T5 responses under visual stimulation and suggesting that coding is again more distributed at this stage.

Based on connectomics, one can speculate about candidates for the implementation of these parallel circuit motifs between L1 and Mi1 and Tm3. The lamina neuron L5 and the GABAergic feedback neurons C2 and C3 receive L1 inputs and could be part of an interconnected local microcircuit (*Takemura et al., 2013*). Intercolumnar neurons, not present in the current connectome datasets, like Pm or Dm neurons, might also be involved and are likely glutamatergic (*Davis et al., 2018*; *Raghu and Borst, 2011*). In fact, there are close to 100 cell types in the visual system and ~60 medulla neurons, but their role is so far unknown. Sensory pathway splits in the periphery are one of the most fundamental steps in sensory processing. Turning this into a process that parallel pathways can achieve might make this important feature extraction step robust to perturbations.

## GluClα mutants reveal GABAergic inhibition onto T4 dendrites

T4 flash responses in a *GluClα*-deficient background show an increase in calcium signal during the OFF epoch and a decrease during the ON epoch (*Figure 8*). For a long time, the mechanisms that generate direction-selective responses in T4/T5 neurons were thought to rely on feedforward excitatory mechanisms (*Fisher et al., 2015b*; *Silies et al., 2014*; *Yang and Clandinin, 2018*). Recently, it was suggested that these direction-selective cells in the fly visual system also implement mechanisms that rely on null-direction suppression (*Haag et al., 2016*; *Leong et al., 2016*; *Salazar-Gatzimas et al., 2016*). Whereas electrophysiological recordings showed inhibition in T4 when the trailing edge of the receptive field was specifically stimulated (*Gruntman et al., 2018*), whole-cell recording experiments of T4/T5 neurons are daunting and this is the first time that calcium imaging data directly reveals inhibition in response to single ON flashes. Since glutamatergic inhibition via GluClα was disrupted in this experimental context, our data suggests that this is due to GABAergic inhibition. Several neuronal candidates could make inhibitory synapses onto T4 dendrites. Based on connectomics and neurotransmitter identity, neurons like Mi4, C3, CT1 or TmY15 give direct input and are GABAergic (*Meier and Borst, 2019*; *Takemura et al., 2017*). Alternatively, this decrease in calcium signal in T4 might come from a lack of excitatory inputs in a GluClα mutant background. Interestingly, Mi1 and Tm3 themselves show inhibition in response to light when GluClα is blocked. However, this effect is more pronounced at their dendrites than in their output layer and shows different kinetics. Our work might thus help uncover a GABAergic inhibitory input to T4 that is more strongly apparent in the absence of Mi1 and Tm3 excitation, and could ultimately reveal the circuit implementation for the inhibitory component of T4/T5 receptive fields (*Leong et al., 2016*; *Salazar-Gatzimas et al., 2016*). Furthermore, our data also reveals an increase in calcium during OFF stimulation. The major inputs to T4 are themselves rectified (*Behnia et al., 2014*). However, rectification in T4 might not be purely inherited by its inputs but also further strengthened at the T4 dendrites. Our findings thus suggest that glutamatergic inhibition contributes to establishing or maintaining contrast selectivity in T4.

## Evolutionary advantages of glutamatergic ionotropic inhibition

Both *GluClα* and *Rdl* are ionotropic ligand-gated receptors. While ionotropic receptors also implement the ON and OFF pathway split in *C. elegans* chemosensation (*Chalasani et al., 2007*), examples in vertebrate vision, olfaction and gustation require metabotropic receptors (*Chandrashekar et al., 2006*; *Masu et al., 1995*; *Nei et al., 2008*). Ionotropic receptors appear to be more common in insects than in vertebrates (*Silbering and Benton, 2010*). Furthermore, glutamate-gated chloride channels have independently arisen three times within invertebrate clades and are present in arthropods, molluscs and flatworms (*Lynagh et al., 2015*), arguing for a strong evolutionary benefit. Ionotropic receptors mediate rapid transduction events at scales smaller than a

millisecond, whereas metabotropic ones are in the millisecond to second range and last longer, from seconds to several minutes, due to an enzymatic secondary cascade previous to channel opening (*Betz, 1990*; *Shiells, 1994*). The evolutionary choice of the specific glutamatergic inhibitory system needs to match the sensory processing speed required for accurate behavioral responses in these species. For example, at the photoreceptor level, invertebrate phototransduction is faster than vertebrate phototransduction thanks to sophisticated molecular strategies (*Hardie and Raghu, 2001*; *Katz and Minke, 2009*). Also, the latency of olfactory sensory neurons responses in mammals is longer than that observed in insects (*Sato et al., 2008*; *Silbering and Benton, 2010*). One advantage that metabotropic receptors have over ionotropic receptors is further amplification of the signal (*Shiells, 1994*). The distributed circuit architecture proposed here might therefore strengthen signaling in a system that uses ionotropic signaling.

### Potential implications for sensory processing in other systems

Here we showed that ON selectivity is not a monosynaptic process as described in other systems (*Chalasani et al., 2007*; *Masu et al., 1995*). Although acute pharmacological block or a systemic loss of function of GluClα abolished all ON responses in different neurons, cell-type-specific mutants retained intact ON responses, revealing that sensory coding is distributed in the fly visual system. This not only highlights the power of fly genetics but sheds new light onto the mechanisms of ON selectivity in other systems, since conclusions about ON and OFF pathway splits being mediated by specific monosynaptic processes in systems such as the vertebrate retina or the *C. elegans* chemosensory system relied on systemic loss-of-function approaches (*Chalasani et al., 2007*; *Masu et al., 1995*). Several of these systems allow for cell-type-specific manipulations using genetic approaches. It will be interesting to revisit these systems and ask if coding is similarly distributed across multiple synapses in different sensory systems and organisms.

# Materials and methods

**Key resources table**

| Reagent type (species) or resource | Designation | Source or reference | Identifiers | Additional information |
|---|---|---|---|---|
| Strain, strain background (*Drosophila melanogaster*) | *Mi1 >> GCaMP6f* | Bloomington Drosophila Stock Center | w+; R19F01-p65ADZp$^{attP40}$ / +; R71D01-ZpGdbd$^{attP2}$/UAS-GCaMP6f | *Figures 1, 2* and *7 Figure 2—figure supplement 1* |
| Strain, strain background (*Drosophila melanogaster*) | *Tm3 >> GCaMP6f* | Bloomington Drosophila Stock Center | w+; R38C11-p65ADZp$^{attP40}$ / +; R59C10-ZpGdbd$^{attP2}$/ UAS- GCaMP6f | *Figures 1, 2* and *7 Figure 2—figure supplement 1* |
| Strain, strain background (*Drosophila melanogaster*) | *L1 >> GCaMP6f* | Bloomington Drosophila Stock Center | w$^+$; L1[c202]-Gal4 / +; UAS-GCaMP6f / + | *Figure 1, 2* |
| Strain, strain background (*Drosophila melanogaster*) | *Mi1 >> iGluSnFR* | Bloomington Drosophila Stock Center | w+; R19F01-p65ADZp$^{attP40}$ / +; R71D01-ZpGdbd$^{attP2}$/UAS iGluSnFR A184A $^{attP2}$ | *Figure 1, Figure 2—figure supplement 3* |
| Strain, strain background (*Drosophila melanogaster*) | *Tm3 >> iGluSnFR* | Bloomington Drosophila Stock Center | w+; R38C11-p65ADZp[attP40] / +; R59C10-ZpGdbd$^{attP2}$/UAS iGluSnFR A184A$^{attP2}$ | *Figure 1, Figure 2—figure supplement 3* |

*Continued on next page*

*Continued*

| Reagent type (species) or resource | Designation | Source or reference | Identifiers | Additional information |
|---|---|---|---|---|
| Strain, strain background (*Drosophila melanogaster*) | T4/T5 >> GCaMP6f | Bloomington Drosophila Stock Center | $w^+$; R64G09-LexA$^{attP40}$, lexAop2-IVS-GCaMP6f-p10$^{su(Hw)attP5}$ / +; + / + | *Figure 2, Figure 8* |
| Strain, strain background (*Drosophila melanogaster*) | GluClα $^{MI02890-GFSTF.2}$ | Bloomington Drosophila Stock Center | $y^1$ $w^*$; Mi{PT-GFSTF.2} GluClα $^{MI02890-GFSTF.2}$/TM6C, Sb$^1$ Tb$^1$ | *Figure 3* |
| Antibody | Anti-GFP (chicken polyclonal) | Abcam | Cat# ab13970, RRID:AB_300798 | IF (1:2000) *Figure 3, Figure 4—figure supplement 2* |
| Antibody | Anti-Bruchpilot (mouse monoclonal nc82) | DSHB | Cat# nc82, RRID:AB_2314866 | IF (1:25) *Figure 3* |
| Antibody | Alexa Fluor 488-conjugates AffinityPure Goat Anti-Chicken IgG | Jackson Immuno Research Labs | Cat# 103-545-155, RRID:AB_2337390 | IF (1:200) *Figure 3* |
| Antibody | Alexa Fluor 594-conjugates AffinityPure Goat Anti-Mouse IgG | Jackson Immuno Research Labs | Cat# 115-585-206, RRID:AB_2338886 | IF (1:200) *Figure 3* |
| Strain, strain background (*Drosophila melanogaster*) | Mi1 >> GCaMP6f, Rdl$^1$/+ control | Bloomington Drosophila Stock Center | $w^+$; R19F01-LexA$^{attP40}$, lexAop2-IVS-GCaMP6f-p10$^{su(Hw)attP5}$ / +; Rdl$^1$ / + | *Figure 4, Figure 4—figure supplement 1* |
| Strain, strain background (*Drosophila melanogaster*) | Tm3 >> GCaMP6f, +/+ control | Bloomington Drosophila Stock Center | $w^+$; R13E12-LexA$^{attP40}$, lexAop2-IVS-GCaMP6f-p10$^{su(Hw)attP5}$ / +; + / + | *Figure 4, Figure 4—figure supplement 1* |
| Strain, strain background (*Drosophila melanogaster*) | Mi1 >> GCaMP6f, Rdl$^1$/RdlMD$^{MD-RR}$ * | Bloomington Drosophila Stock Center | $w^+$; R19F01-LexA$^{attP40}$, lexAop2-IVS-GCaMP6f-p10$^{su(Hw)attP5}$ / +; Rdl$^1$/RdlMD$^{MD-RR}$ | *Figure 4, Figure 4—figure supplement 1* |
| Strain, strain background (*Drosophila melanogaster*) | Tm3 >> GCaMP6f, Rdl$^{MD-RR}$/RdlMD$^{MD-RR}$ * | Bloomington Drosophila Stock Center | $w^+$; R13E12-LexA$^{attP40}$, lexAop2-IVS-GCaMP6f-p10$^{su(Hw)attP5}$ / +; Rdl$^{MD-RR}$/RdlMD$^{MD-RR}$ | *Figure 4, Figure 4—figure supplement 1* |
| Strain, strain background (*Drosophila melanogaster*) | C2,C3 >> GFP | Bloomington Drosophila Stock Center | w+/UAS-CD8::GFP; R20C11-p65ADZp$^{attP40}$/UAS-2xEGFP; R48D11-ZpGdbd$^{attP2}$/+ | *Figure 4—figure supplement 2* |
| Strain, strain background (*Drosophila melanogaster*) | L1 >> GFP | Bloomington Drosophila Stock Center | $w^+$/UAS-CD8::GFP; L1[c202]-Gal4/UAS-2xEGFP; + / + | *Figure 4—figure supplement 2* |
| Antibody | Anti-GABA (rabbit polyclonal) | Sigma-Aldrich | Cat# A2052, RRID:AB_477652 | IF (1:200) *Figure 4—figure supplement 2* |
| Antibody | Alexa Fluor 594-conjugates AffiniPure Goat Anti-Rabbit IgG | Jackson Immuno Research Labs | Cat# 111-585-003, RRID:AB_2338059 | IF (1:200) *Figure 6* |
| Strain, strain background (*Drosophila melanogaster*) | Mi1 >> GCaMP6f, GluClα$^{Df}$/+ control | Bloomington Drosophila Stock Center | $w^+$; R19F01-LexA$^{attP40}$, lexAop2-IVS-GCaMP6f-p10$^{su(Hw)attP5}$ / +; Df(3R)ED6025/ + | *Figure 6, Figure 6—figure supplement 1* |

*Continued on next page*

Continued

| Reagent type (species) or resource | Designation | Source or reference | Identifiers | Additional information |
|---|---|---|---|---|
| Strain, strain background (*Drosophila melanogaster*) | *Tm3 >> GCaMP6f, GluClα$^{Df}$/+ control* | Bloomington Drosophila Stock Center | *w $^+$; R13E12-LexA$^{attP40}$, lexAop2-IVS-GCaMP6f-p10$^{su(Hw)attP5}$ / +; Df(3R)ED6025/ +* | *Figure 6, Figure 6—figure supplement 1* |
| Strain, strain background (*Drosophila melanogaster*) | *Mi1 >> GCaMP6f, GluClα$^{S278T}$/GluClα$^{Df}$* | This paper | *w $^+$; R19F01-LexA$^{attP40}$, lexAop2-IVS-GCaMP6f-p10$^{su(Hw)attP5}$ / +; Df(3R)ED6025/GluClα$^{S278T}$* | *Figure 6, Figure 6—figure supplement 1 More information in the Materials and methods section under 'Molecular biology'* |
| Strain, strain background (*Drosophila melanogaster*) | *Tm3 >> GCaMP6f, GluClα$^{S278T}$/GluClα$^{Df}$* | This paper | *w $^+$; R13E12-LexA$^{attP40}$, lexAop2-IVS-GCaMP6f-p10$^{su(Hw)attP5}$ / +; ; Df(3R)ED6025/ GluClα$^{S278T}$* | *Figure 6, Figure 6—figure supplement 1 More information in the Materials and methods section under 'Molecular biology'* |
| Strain, strain background (*Drosophila melanogaster*) | *T4/T5 >> GCaMP6f, GluClα$^{Df}$/+ control* | Bloomington Drosophila Stock Center | *w$^+$; R64G09-LexA$^{attP40}$, lexAop2-IVS-GCaMP6f-p10$^{su(Hw)attP5}$ / +; ; Df(3R)ED6025 / +* | *Figure 6, Figure 6—figure supplement 1* |
| Strain, strain background (*Drosophila melanogaster*) | *T4/T5 >> GCaMP6f, GluClα$^{S278T}$/GluClα$^{Df}$* | This paper | *w$^+$; R64G09-LexA$^{attP40}$, lexAop2-IVS-GCaMP6f-p10$^{su(Hw)attP5}$ / + ; Df(3R)ED6025/ GluClα$^{S278T\ CRISPR}$* | *Figure 6, Figure 6—figure supplement 1 More information in the Materials and methods section under 'Molecular biology'* |
| Strain, strain background (*Drosophila melanogaster*) | *Mi1 >> Flp,GCaMP6f; GluClα$^{FlpStop.ND}$/GluClα$^{Df}$* | This paper | *w+; R19F01-p65ADZp$^{attP40}$/UAS-GCaMP6f, UAS-Flp; R71D01-ZpGdbd$^{attP2}$, GluClα$^{Df}$/GluClα$^{FlpStop.ND}$* | *Figure 7 More information in the Materials and methods section under 'Generation of transgenic lines'* |
| Strain, strain background (*Drosophila melanogaster*) | *Mi1 >> Flp,GCaMP6f; GluClα$^{FlpStop.ND}$ / +(Heterozygous control)* | This paper | *w+; R19F01-p65ADZp$^{attP40}$/UAS-GCaMP6f, UAS-Flp; R71D01-ZpGdbd$^{attP2}$/GluClα$^{FlpStop.ND}$* | *Figure 7 More information in the Materials and methods section under 'Generation of transgenic lines'* |
| Strain, strain background (*Drosophila melanogaster*) | *Mi1 >> GCaMP6f; GluClα$^{FlpStop.ND}$/GluClα$^{Df}$ (No Flp control)* | This paper | *w+; R19F01-p65ADZp$^{attP40}$/UAS-GCaMP6f; R71D01-ZpGdbd$^{attP2}$, GluClα$^{Df}$ / ; GluClα$^{FlpStop.ND}$* | *Figure 7 More information in the Materials and methods section under 'Generation of transgenic lines'* |
| Strain, strain background (*Drosophila melanogaster*) | *Mi1 >> GCaMP6f, GluClα$^{dsRNA}$* | Bloomington Drosophila Stock Center | *w+; R19F01-p65ADZp$^{attP40}$/P{y[+t7.7] v[+t1.8]=TRiP.HMC03585}$^{attP40}$; R71D01-ZpGdbd$^{attP2}$/UAS-GCaMP6f* | *Figure 7* |
| Strain, strain background (*Drosophila melanogaster*) | *Tm3 >> GCaMP6f, GluClα$^{dsRNA}$* | Bloomington Drosophila Stock Center | *w+; R38C11-p65ADZp$^{attP40}$/P{y[+t7.7] v[+t1.8]=TRiP.HMC03585}$^{attP40}$; R59C10-ZpGdbd$^{attP2}$/ UAS-GCaMP6f* | *Figure 7* |

*Continued*

| Reagent type (species) or resource | Designation | Source or reference | Identifiers | Additional information |
|---|---|---|---|---|
| Strain, strain background (*Drosophila melanogaster*) | *Mi1 >> GCaMP6f, GluClα<sup>FlpStop.D</sup> / +* | This paper | *w <sup>+</sup>; R19F01-LexA<sup>attP40</sup>, lexAop2-IVS-GCaMP6f-p10<sup>su(Hw)attP5</sup> / +; GluClα<sup>FlpStop.D</sup> / +* | *Figure 7 More information in the Materials and methods section under 'Generation of transgenic lines'* |
| Strain, strain background (*Drosophila melanogaster*) | *Mi1 >> GCaMP6f, GluClα<sup>Df</sup> / +* | Bloomington Drosophila Stock Center | *w <sup>+</sup>; R19F01-LexA<sup>attP40</sup>, lexAop2-IVS-GCaMP6f-p10<sup>su(Hw)attP5</sup> / +; ; Df(3R)ED6025/GluClα<sup>WT</sup>* | *Figure 7* |
| Strain, strain background (*Drosophila melanogaster*) | *Mi1 >> GCaMP6f, GluClα<sup>FlpStop.D</sup>/GluClα<sup>Df</sup> \*\** | This paper | *w <sup>+</sup>; R19F01-LexA}<sup>attP40</sup>, lexAop2-IVS-GCaMP6f-p10<sup>su(Hw)attP5</sup> / +; ; Df(3R)ED6025/GluClα<sup>FlpStop.D</sup>* | *Figure 7 More information in the Materials and methods section under 'Generation of transgenic lines'* |
| Strain, strain background (*Drosophila melanogaster*) | *Tm3 >> GCaMP6f, GluClα<sup>FlpStop.D</sup> / +* | This paper | *w <sup>+</sup>; R13E12-LexA<sup>attP40</sup>, lexAop2-IVS-GCaMP6f-p10<sup>su(Hw)attP5</sup> / +; GluClα<sup>FlpStop.D</sup> / +<sup>T</sup>* | *Figure 7 More information in the Materials and methods section under 'Generation of transgenic lines'* |
| Strain, strain background (*Drosophila melanogaster*) | *Tm3 >> GCaMP6f, GluClα<sup>Df</sup> / +* | Bloomington Drosophila Stock Center | *w <sup>+</sup>; R13E12-lexA}<sup>attP40</sup>, lexAop2-IVS-GCaMP6f-p10<sup>su(Hw)attP5</sup> / +; ; Df(3R)ED6025/ +* | *Figure 7* |
| Strain, strain background (*Drosophila melanogaster*) | *Tm3 >> GCaMP6f, GluClα<sup>FlpStop.D</sup>/GluClα<sup>Df</sup> \*\** | This paper | *w <sup>+</sup>; R13E12-LexA<sup>attP40</sup>, lexAop2-IVS-GCaMP6f-p10<sup>su(Hw)attP5</sup> / +; ; Df(3R)ED6025/GluClα<sup>FlpStop.D</sup>* | *Figure 7 More information in the Materials and methods section under 'Generation of transgenic lines'* |
| Strain, strain background (*Drosophila melanogaster*) | *T4/T5 >> GCaMP6f, GluClα<sup>FlpStop.D</sup> / +* | This paper | *w<sup>+</sup>; R64G09-LexA<sup>attP40</sup>, lexAop2-IVS-GCaMP6f-p10<sup>su(Hw)attP5</sup> / +; GluClα<sup>FlpStop.D</sup> / +* | *Figure 8 More information in the Materials and methods section under 'Generation of transgenic lines'* |
| Strain, strain background (*Drosophila melanogaster*) | *T4/T5 >> GCaMP6f, GluClα<sup>Df</sup> / +* | Bloomington Drosophila Stock Center | *w<sup>+</sup>; R64G09-LexA<sup>attP40</sup>, lexAop2-IVS-GCaMP6f-p10<sup>su(Hw)attP5</sup> / +; ; Df(3R)ED6025/ +* | *Figure 8* |
| Strain, strain background (*Drosophila melanogaster*) | *T4/T5 >> GCaMP6f, GluClα<sup>FlpStop.D</sup>/GluClα<sup>Df</sup> \*\** | This paper | *w<sup>+</sup>; R64G09-LexA<sup>attP40</sup>, lexAop2-IVS-GCaMP6f-p10<sup>su(Hw)attP5</sup> / +; ; Df(3R)ED6025/GluClα<sup>FlpStop.D</sup>* | *Figure 8 More information in the Materials and methods section under 'Generation of transgenic lines'* |
| Chemical compound, drug | *Picrotoxin* | Sigma Aldrich | P1675_SIGMA | *Figures 2, 4, 5 and 6 Figure 2—figure supplements 2 and 3, Figure 4—figure supplement 1, Figure 6—figure supplement 1* |
| Chemical compound, drug | *MPEP* | Abcam | Ab120008 | *Figure 2—figure supplement 1* |

*Continued on next page*

*Continued*

| Reagent type (species) or resource | Designation | Source or reference | Identifiers | Additional information |
|---|---|---|---|---|
| Sequence-based reagent | GluCla_forward | This paper | ACCAAACTGC TGCAAGAC | qRT-PCR *Figure 7* *More information in the Materials and methods section under 'Molecular biology'* |
| Sequence-based reagent | GluCla_reverse | This paper | GATATGTGCTCC AGTAGACC | qRT-PCR *Figure 7* *More information in the Materials and methods section under 'Molecular biology'* |
| Sequence-based reagent | GAPDH2_forward | This paper | GATGAGGAGGT CGTTTCTAC | qRT-PCR *Figure 7* *More information in the Materials and methods section under 'Molecular biology'* |
| Sequence-based reagent | GAPDH2_reverse | This paper | GTACTTGATC AGGTCGATG | qRT-PCR *Figure 7* *More information in the Materials and methods section under 'Molecular biology'* |
| Software, algorithm | MATLAB R2017a | The MathWorks Inc.50 Natick, MA | Custom scripts | *Codes are available in the* **Source code 1** |

*We used different allelic combinations for the Rdl$^{MDRR}$ insensitive allele when imaging Mi1 (*Rdl$^{MD-RR}$/Rdl$^1$*) or Tm3 (*Rdl$^{MD-RR}$/RdlMD$^{MD-RR}$*). While the use of the Rdl$^1$ null mutant is genetically cleaner, application of low concentrations of PTX has weaker phenotypes in genetic backgrounds carrying the *Rdl$^1$* allele than in wild type, possibly due to homeostatic mechanisms (**Figure 2A,B**, **Figure 4A**). The 2.5 µM PTX phenotype was even weaker in Tm3, and did not leave a margin to look for rescue by *Rdl$^{MDRR}$*, which is why we instead used two copies of the *Rdl$^{MDRR}$* allele, which has a PTX phenotype similar to wild type in heterozygosity.

**GluCla$^{FlpStop.D}$/Df* mutant larvae failed to crawl out of the food, but adult flies could be obtained after saving pupae from the food.

## *Drosophila* strains and fly husbandry

Flies were raised at 25℃ and 55% humidity on molasses-based food on a 12:12 hr light:dark cycle. Imaging experiments were done at room temperature (20℃). Genotypes of all *Drosophila* strains used for experiments are listed in the Key Resources Table.

## Immunohistochemistry and confocal microscopy

Female flies were dissected 3–5 days after eclosion. Brains were removed in dissection solution and fixed in 2% paraformaldehyde in phosphate buffered lysine (PBL) for 50 min at room temperature. Subsequently, the brains were washed 3x for 5 min in phosphate buffered saline containing 0.3% Triton X-100 (PBT) adjusted to pH 7.2. For antibody staining, the samples were blocked in 10% normal goat serum (NGS, Fisher Scientific GmbH, Schwerte, Germany) in PBT for 30 min at room temperature followed by incubation for 24 hr at 4℃ in the primary antibody solution (mouse mAb nc82,1:25, DSHB; chicken anti-GFP,1:2000, Abcam ab13970; rabbit anti-GABA, 1:200, Sigma-Aldrich, A2052). Primary antibodies were removed by washing in PBT 3 times for 5 min and the brains were incubated in the secondary antibody (anti-chicken-Alexa488, anti-mouse-Alexa594, anti-rabbit-Alexa594, all 1:200, Dianova) in the dark at 4℃ overnight. The samples were further washed with PBT (3 × 5 min) and mounted in Vectashield (Vector Laboratories, Burlingame).

Serial optical sections were taken on a Zeiss LSM710 microscope (Carl Zeiss Microscopy GmbH, Germany) equipped with an oil immersion Plan-Apochromat 40x (NA = 1.3) objective and using the Zen 2 Blue Edition software (Carl Zeiss Microscopy, LLC, United States). Z-stack images were taken at 1 µm intervals and 512 × 512 pixel resolution. Confocal stacks were rendered into two-

dimensional images using Fiji (*Schindelin et al., 2012*). The images were then further processed using Illustrator CS5.1 (Adobe) or Inkscape version 0.92.1 (The Inkscape Team).

## Generation of transgenic lines

Transgenic lines carrying the FlpStop cassette (*Fisher et al., 2017*) for conditional gene control were generated according to standard procedures. In brief, embryos carrying the Mi02890 insertion (*y[1] w[*]; Mi{y[+mDint2]=MIC}GluClα^MI02890^/TM3,Sb^1^*) were injected with the FlpStop cassette and PhiC31 integrase. Embryos were dechorionated in 50% bleach (DanKlorix) for 3 min, followed by washing in a buffer (100 mM NaCl, 0.02% Triton X-100) for 3 min. Injections were done on a Nikon AZ100 microscope using a FemtoJet 4i (Eppendorf AG, Hamburg, Germany). The injection mix (20 µl) consisted of 10 µg of the FlpStop construct, 6 µg of helper DNA (pBS130 containing the PhiC31 integrase) and 4 µl of 5x injection buffer (25 mM KCl, 0.5 mM $NaH_2PO4$, pH 6.8, 1% phenol red [Sigma Aldrich]). Injection needles were pulled from quartz glass microcapillaries (10 cm length, 1.0 mm outside diameter, 0.5 mm inside diameter, Sutter Instruments, USA) using a P-2000 micropipette puller (Sutter Instruments, USA). Needles were sharpened using a capillary grinder (Bachofer, Germany). After injection, embryos were covered with 10S Voltalef oil and incubated at 18°C until larval hatching. Successful recombinase-mediated cassette exchange was scored by the loss of the yellow marker (y[+]) of the MiMIC cassette and verified by single fly PCR, testing for the loss of the MiMIC cassette and the orientation of the inserted FlpSTOP cassette, as in *Fisher et al. (2017)*.

## Molecular biology

### GluClα PTX-insensitive allele

To generate a PTX-insensitive GluClα allele, (seamless), CRISPR/Cas9-based genome editing was used to introduce the S278T mutation. Mutagenesis was performed by Well genetics (Taiwan) using the following guide RNA (gRNA): ATCATGGGTATCATTCTGGC[TGG]. In brief, the gRNA was cloned into a U6 promoter plasmid. Cassette-inverted PBacDsRed containing two PBac terminals, 3xP3-DsRed and two homology arms with point mutation S278T was cloned into pUC57-Kan as donor template for repair. GluClα-targeting gRNAs and hs-Cas9 were supplied in DNA plasmids, together with donor plasmid for microinjection into embryos of control strain w1118. F1 flies carrying the selection marker 3xP3-DsRed were validated by genomic PCR and sequencing. Subsequently, the 3x-P3 dsRed selection marker was removed using tub-PBac\T transposase (BDSC# 8285), and successful mutagenesis and 3xP3-dsRed removal was confirmed by PCR and sequencing.

### qRT-PCR

For RNA extraction, ten adult fly brains per biological replicate were dissected in PBS. RNA extraction was performed using the RNeasy Mini Kit (Qiagen). The RNA quality was confirmed by the presence of both 18S and 28S ribosomal peaks and the lack of evidence of RNA degradation (RNA integrity number >5) (Fragment Analyser, Agilent). cDNA synthesis was done using SuperScript VILO kit and master mix (Thermo Fisher Scientific). For the qRT-PCR, three biological and three technical replicates were used per genotype. For each sample, 1 µl of cDNA (1:8 dilution) was mixed with 7 µl $H_2O$, 10 µl SYBR Green MM and 1 µl forward and 1 µl reverse primers for a given gene. Standard curves for every primer pair were taken at least once in each experiment. For run-to-run variations, each 96-well plate contained positive and negative controls in the same chosen dilution as the experimental samples. The qRT-PCR analysis software (Light Cycler 480) determined the Ct values for each technical replicate using the first peak of the second derivative method. The mean of three readings was used to estimate Ct values for each biological replicate. The Delta-Delta Ct method was used to calculate relative transcript levels: % Transcript difference = $2^{\wedge}$([Ct ND target gene – Ct ND reference gene] – [Ct D target gene – Ct D reference gene]). For presentation purposes, all observations were normalized to the mean transcript level of the WT genotype. The housekeeping gene *GAPDH2* was used as reference gene for relative mRNA quantification of *GluClα* levels. Primers were designed to amplify a product spanning the two exons flanking the intron containing the FlpStop cassette, *GluClα* exons 18 and 19, resulting in a 137 bp long amplicon. All primers were tested for primer efficiency using serial dilutions of the WT control, and efficiency values are noted next to each primer pair below.

GluCla_forward: ACCAAACTGCTGCAAGAC

GluCla_reverse: GATATGTGCTCCAGTAGACC
(Efficiency: 1.91)
GAPDH2_forward: GATGAGGAGGTCGTTTCTAC
GAPDH2_reverse: GTACTTGATCAGGTCGATG
(Efficiency: 1.96)

## RNA-seq

Raw sequencing reads and TPM tables from published datasets were taken from (*Konstantinides et al., 2018*; GSE 103772) and (*Davis et al., 2018*; GSE 116969). To estimate transcript abundance, we used Kallisto (v0.43.1; *Bray et al., 2016*) to pseudo-align reads to dm6 annotation (ENSEMBLE release 91 derived from FlyBase release version 2017_04). The TPM matrix was processed further in R studio (R version 3.4.4). The TPMs were summarized at the level of genes averaged across cell type replicates. Heat maps of the gene expression in selected cell types was generated using MultiExperiment Viewer (MeV) 4.9.0 (*Howe et al., 2011*).

## In vivo two-photon imaging

### Fly preparation, experimental setup and data acquisition

Female flies aged 2 to 5 days were used for calcium imaging experiments, with the exception of *GluClα* cell-type-specific disruption experiments in which the flies were 8 to 10 days old. Flies were anesthetized on ice and then glued with a UV-sensitive glue (Bondic) onto a custom-made microscope holder containing a hole fitting head and thorax of the fly. To expose the brain, the cuticle on the back of the head was removed using breakable razor blades and fine forceps. During imaging, the flies were perfused with a carboxygenated saline containing 103 mM NaCl, 3 mM KCl, 5 mM TES, 1 mM NaH$_2$PO$_4$, 4 mM MgCl$_2$, 1.5 mM CaCl$_2$, 10 mM trehalose, 10 mM glucose, 7 mM sucrose, and 26 mM NaHCO$_3$. The pH of the saline equilibrated near 7.3 when bubbled with 95% O$_2$/5% CO$_2$. Imaging experiments were performed on a Bruker Investigator two-photon microscope (Bruker, Madison, WI, USA), equipped with a 25x/1.1 objective (Nikon, Minato, Japan). The excitation laser (Spectraphysics Insight DS+) was set to 920 nm in order to excite GCaMP6f, applying 5–15 mW of power to the sample. Emitted light was sent through an SP680 short pass filter, a 560 lpxr dichroic filter and a 525/70 emission filter. Data was acquired using the PrairieView software at a frame rate of ~10–15 Hz and around 3-5x zoom depending on the cell type.

### Pharmacology

All pharmacological agents were stored, handled, and disposed of as indicated in the corresponding SDS and/or information sheets. MPEP was purchased from Abcam (ab120008) and PTX from Sigma Aldrich (R284556). All toxins were first dissolved in water and kept as concentrated stock solutions at −20°C for a maximum of 2 months. For experiments, toxin stock solutions were allowed to equilibrate to room temperature and diluted to the appropriate concentrations in the calcium imaging solution. This solution was then used for a maximum of 3 days, stored at 4°C. All toxins were bath applied. No perfusion was used before or after toxin application. The toxin was allowed to penetrate for 10 min before the start of experiments. The same regions of interest (ROIs) were imaged before and after toxin application to allow paired comparisons for toxin effects.

### Visual stimulation

Visual stimuli were generated using custom-written software using C++ and OpenGL, and presented using a LightCrafter 4500 (Texas Instruments, Texas, USA) running at a frame rate of 100 Hz. The imaging and the visual stimulus presentation were synchronized as described in *Freifeld et al. (2013)*. Stimulus light was filtered with a 482/18 bandpass and ND1.0 neutral density filter and projected onto an 8 cm x 8 cm rear projection screen positioned in front of the fly and spanning a visual angle of 60 in azimuth and elevation.

### Periodic full-field flashes

Periodic, alternating full-contrast ON and OFF flashes covering the whole screen, each lasting 5 s, were presented to the flies. Each stimulus epoch was presented for ~7 trials.

### Moving OFF and ON edges

The stimulus consisted of 100% contrast moving bright or dark edges moving at a velocity of 20°/s in eight directions covering 360°. All stimuli were presented in random order with at least three repetitions per stimulus.

## Data analysis

All data processing was performed offline using MATLAB R2017a (The MathWorks Inc, Natick, MA). To correct for motion artifact, individual images were aligned to a reference image composed of a maximum intensity projection of the first 30 frames. The average intensity for manually selected ROIs was computed for each imaging frame and background subtracted to generate a time-trace of the response. ROI identities were kept for matching identical ROIs before and after toxin application for paired analysis. All responses and visual stimuli were interpolated at 10 Hz and trial averaged. Neural responses are shown as relative fluorescence intensity changes over time ($\Delta F/F_0$). The mean and the standard error of the mean (SEM) were calculated across flies after averaging over ROIs for each fly. A two-tailed Student $t$ test for paired or unpaired (independent) samples was used for statistical analysis between two groups. For comparisons between more than two groups in which one independent variable was manipulated (here: PTX concentrations), one-way ANOVA followed by two-tailed t-test with Bonferroni-Holm correction for multiple comparisons was used. For multiple comparisons between groups in which two independent variables were manipulated (here: PTX concentration and genotype), an unbalanced two-way ANOVA followed by Tukey's Honestly Significant Difference Procedure for multiple comparisons was used. For all the data, normality was tested with a Lilliefors test.

### Full-field flashes

To calculate $\Delta F/F_0$, the mean of the whole trace was used as $F_0$. Step responses were calculated as the difference between the mean response 500 ms before the onset of the stimulus and the peak $\Delta F/F_0$ during the stimulus epoch. Plateau responses were calculated as the difference between the mean response 500 ms before the onset of the stimulus and the mean of the last 500 ms of the stimulus epoch. Integrated responses were calculated as the sum of all values during the 5 s of the ON stimulus epoch. For Mi1 and Tm3, only ROI responses that were positively correlated with the stimulus (Pearson's Correlation Coefficient >0) were used for subsequent analysis (see also *Fisher et al., 2015a*). To discard noisy ROIs, a standard deviation threshold of 0.2 was set during the 2 s before the onset of visual stimulation. For pharmacological experiments, an absolute threshold of 0.5 $\Delta F/F_0$ was used to identify well-responding ROIs. The identical ROIs were analyzed after toxin application (paired data). When control and experimental conditions were not paired, no threshold was applied. When experimental (and respective controls) conditions in preliminary data showed a full loss of response, no absolute threshold was applied and the recordings were obtained blinded to the experimenter. For T4/T5 single ROI responses, no correlation filter was applied since both response polarities were expected. For iGluSnFR recordings, only the response threshold was applied.

### Moving ON and OFF edges

To calculate $\Delta F/F_0$, the mean of the last second of the intermediate gray epoch was used as $F_0$. ROI trial average was done per epoch (one specific direction of movement and contrast change). The mean response across ROIs was calculated after aligning traces by their maximum responses. Response amplitudes were calculated as the difference between the mean response 500 ms before the onset of the stimulus and the maximum during the stimulus epoch. The direction selectivity index was calculated as PD-ND/PD, where PD is the maximum response among all responses to the different directions of motion and ND is the response to the null direction of motion defined as 180° from the preferred direction.

## Electrophysiological recordings of heterologously expressed GluClα constructs

For electrophysiological recordings, GluClα constructs (isoform O, NP_001287409) were heterologously expressed in *Xenopus laevis* oocytes. Oocytes were harvested from our own colony. Frogs were housed according to the German law of animal protection and the district veterinary office.

Oocytes were harvested following standard procedures and in agreement with the animal testing approval 84–02.04.2016.A077.

GluClα constructs were cloned in the pGEMHE vector. The vector was linearized with NheI and transcribed using the T7 mMessage mMachine kit (Ambion, Austin, TX). *Xenopus* oocytes were injected with 50 nl RNA (0.01–0.2 µg/µl) and incubated at 14–16°C for 1–2 days in ND96 medium containing (in mM): 96 NaCl, 2 KCl, 1.8 CaCl$_2$, 1 MgCl$_2$, 10 4-(2-hydroxyethyl) piperazine-1-ethane-sulfonic acid (HEPES), 5 Na-pyruvate, and 100 mg/l gentamicin, adjusted to pH 7.5 with NaOH.

Electrophysiological experiments were performed at room temperature (22–25°C). Oocytes were placed in an RC-3Z recording chamber (Warner Instruments, Hamden, CT) under a Discovery V8 stereoscope (Zeiss, Oberkochen, Germany) and continuously perfused with ND96 by a PC-controlled gravity-driven system with a flux rate of 7 ml/min. Electrodes were pulled from 1.5 mm thick borosilicate glass capillaries (Hilgenberg, Malsfeld, Germany) on a DMZ puller (Zeitz Instruments GmbH, Martinsried, Germany) and filled with 3 M KCl. The resulting initial electrode resistance was 0.5–5 MΩ in ND96. Currents were recorded in the two-electrode voltage-clamp mode at a holding potential of −70 mV with a Gene Clamp 500 amplifier (Molecular Devices, San Jose, CA), connected via a USB-6341 acquisition board (National Instruments, Austin, TX) to a PC running WinWCP (Strathclyde, University of Glasgow, UK). L-glutamate was dissolved in ND96 and was repeatedly applied for 20 s, with an interstimulus interval of 1–2 min to ensure full recovery from desensitization. Picrotoxin stock solution was first prepared in dimethyl sulfoxide (DMSO) and then diluted in ND96.

The peak current amplitude of the glutamate-evoked response, in the presence of the antagonist, was normalized to the mean peak current amplitude evoked by glutamate after picrotoxin wash-out. Data are shown as the mean ± SD. N indicates the number of cells. Data was analyzed with Igor Pro (Wavemetrics, Portland, OR). An unpaired *t* test was used for statistical analysis of the residual current between WT and the S278T mutant allele.

## Acknowledgements

We are grateful to Jonas Chojetzki for excellent technical assistance and to all members of the Silies lab for helpful discussion. We thank Carlotta Martelli, Carsten Duch, Jan Clemens and Yvette Fisher for comments on the manuscript. This work was supported by the DFG through SFB889 (grant # 154113120), Mechanisms of Sensory Processing, Project C08, and the Emmy Noether Program, grant SI1991/1-1.

## Additional information

### Funding

| Funder | Grant reference number | Author |
| --- | --- | --- |
| Deutsche Forschungsgemeinschaft | Emmy Noether SI 1991/1-1 | Miriam Henning Burak Gür Junaid Akhtar Marion Silies |
| Deutsche Forschungsgemeinschaft | SFB889, Project C08 | Sebastian Molina-Obando Juan Felipe Vargas-Fique |

The funders had no role in study design, data collection and interpretation, or the decision to submit the work for publication.

### Author contributions

Sebastian Molina-Obando, Conceptualization, Data curation, Formal analysis, Investigation, Methodology, Writing—original draft; Juan Felipe Vargas-Fique, Formal analysis, Investigation, Writing—review and editing; Miriam Henning, Burak Gür, Formal analysis, Investigation; T Moritz Schladt, Investigation; Junaid Akhtar, Formal analysis, Writing—review and editing; Thomas K Berger, Conceptualization, Formal analysis, Investigation, Methodology, Writing—original draft; Marion Silies, Conceptualization, Data curation, Supervision, Funding acquisition, Writing—original draft, Project administration

## Author ORCIDs

Sebastian Molina-Obando (iD) http://orcid.org/0000-0003-1222-723X
Burak Gür (iD) http://orcid.org/0000-0001-8221-9767
Marion Silies (iD) https://orcid.org/0000-0003-2810-9828

## Decision letter and Author response

Decision letter https://doi.org/10.7554/eLife.49373.036
Author response https://doi.org/10.7554/eLife.49373.037

## Additional files

### Supplementary files

• Source code 1. Data analysis and statistics.
DOI: https://doi.org/10.7554/eLife.49373.029

• Transparent reporting form DOI: https://doi.org/10.7554/eLife.49373.030

### Data availability

All data generated or analyzed during this study are included in the manuscript and supporting files.

The following previously published datasets were used:

| Author(s) | Year | Dataset title | Dataset URL | Database and Identifier |
|---|---|---|---|---|
| Konstantinides N, Kapuralin K, Fadil C, Barboza L, Satija R, Desplan C | 2018 | RNA sequencing of *Drosophila melanogaster* optic lobe cell types. | https://www.ncbi.nlm.nih.gov/geo/query/acc.cgi?acc=GSE103772 | NCBI Gene Expression Omnibus, GSE103772 |
| Davis FP, Nern A, Picard S, Reiser MB, Rubin GM, Eddy SR, Henry GL | 2019 | A genetic, genomic, and computational resource for exploring neural circuit function | https://www.ncbi.nlm.nih.gov/geo/query/acc.cgi?acc=GSE116969 | NCBI Gene Expression Omnibus, GSE116969 |

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
