## [Decision Letter]

Thank you for submitting your article "ON selectivity in *Drosophila* vision is a multisynaptic process that utilizes both glutamatergic and GABAergic inhibition" for consideration by *eLife*. Your article has been reviewed by three peer reviewers, and the evaluation has been overseen by a Reviewing Editor and Ronald Calabrese as the Senior Editor. The following individuals involved in review of your submission have agreed to reveal their identity: M Eugenia Chiappe (Reviewer #1); Kit Longden (Reviewer #2); Karin Nordström (Reviewer #3).

The reviewers have discussed the reviews with one another and, as the Reviewing Editor, I have drafted this decision to help you prepare a revised submission.

Basically, the three reviewers were very positive about the paper and thought that it should be published. However, there are two important points that you should address before final acceptance of the manuscript:

– One is the writing of the paper. The reviewers made many suggestions as to how to improve it to make it clearer for its audience. It is complex material and you should make every effort to help the readers. I recommend that you look at all the suggestions and come back with a version that addresses these points.

– The other has to do with a point raised by reviewer #1 whether or not L1 cells co-express GABA and Glu. You should provide better evidence to support your point or simply be more careful with your statement on the lack of GABA release from L1 (e.g. subsection “A PTX insensitive GABAAR partially rescues ON responses”).

Otherwise, congratulations on a very elegant paper that will have a significant impact in the field

Reviewer #1:

In this study, Molina-Obando and colleagues have investigated the cellular/circuit mechanisms underlying the emergence of ON selectivity, i.e., the specific sensitivity to increments of light intensity, within circuits of the fly visual medulla. Heretofore, these mechanisms have been poorly understood. Using a clever combination of pharmacology, genetics and physiological tools, this study reveals that: 1) inhibitory signals triggered by glutamate underlie signal transformation, 2) both Gabaergic and Glutamatergic synapses contribute to signal transformation, and 3) although there is a direct inhibitory glutamatergic synapse between presynaptic lamina neurons and the medulla cells under study, activity through this synapse is not critical for ON selectivity. Rather, signal transformation depends on a cascade of multiple synapses that recruit gabaergic- and glutamatergic-based inhibitory activity.

Overall, the authors use a plethora of tools to address specific questions, which gives strength to the observations. However, in some cases, the description of the complex results is not very clear, is incomplete, or poorly discussed. This is important because the observations described in this study strongly suggest that early visual processing is organized in a distributed, multi-synaptic manner, expanding on the original idea of a monosynaptic functional organization. However, the interpretation of the complex results is not trivial, and in some cases, the lack of clarity in the text jeopardises the recognition of the value of the findings.

I think this study is valuable not only for the fly vision community, but also for the broader sensory neuroscience community. The combination of pharmacology with genetic-based channel perturbations in a cell-type specific manner was key to demonstrate the distributed functional organization. Since the separation of signal processing in "ON" vs "OFF" pathways are found in many sensory systems across species, the implication of these results go beyond the fly vision community, and reaches the sensory neuroscience community at large. For these reasons, I would support publication of this work. However, as mentioned above, I found the manuscript a bit difficult to read.

Reviewer #2:

General assessment and major comments

In this manuscript, the authors investigate the molecular mechanisms by which information is transmitted from the contrast encoding laminar monopolar cell L1 to medulla cells Mi1 and Tm3, two of the principal inputs to the direction-selective and ON motion-sensitive T4 cell. Their general strategy initially is to block channels using PTX and rescue function by expressing PTX resistant alleles. They find that low concentrations (5 μM) of PTX block both GABA_A and GluCla receptors (Figure 6). When PTX resistant Rdl alleles are expressed in Mi1 or Tm3, then the steady state response of these cells to ON contrast step changes is rescued (Figure 4). The authors then generate a PTX resistant GluCla allele, for which they are to be commended, and this rescues transient responses to ON contrast step changes (Figure 6). The Tm3 and Mi1 cells are situated in a network of cells including L5 and the GABAergic cells CT1 and C2, so it is entirely credible that L1 signals are polysynaptically communicated to Mi1 and Tm3 cells. In addition, they investigate the activity of T4 and T5 cells under PTX, and in flies with GluCla knocked out.

Overall, I found the manuscript ended well with a great Discussion. The molecular work was impressive and clearly a lot of work. The organization of the results included a lot of implicit reasoning and odd ways of revealing important data that made it difficult to follow. Overall, I was not convinced that establishing a polsynaptic connection between L1 and Tm3 or Mi1 was surprising, and that the paper did not capitalize on this work to understand what the different mechanisms contributed to the ON rectification computation. I think that with a different emphasis, I might have been very impressed with the data. The generation of the PTX-insensitive GluCla receptor allele has the potential to be very useful tool for other circuit dissections. Mainly I have many minor comments, which reflect my enthusiasm for the work.

Major comments

The first sentence of the second paragraph in subsection “ON responses are lost by PTX concentrations affecting GABAARs and GluCls” allow the reader to think that 100 μm PTX would block GABA_AR and GluCla and 5 μm PTX would block GABA_ARs alone. This allows one to think that GABA_Ars are required for normal glutamate signaling under 5 μm PTX in Figure 2. The data in Figure 6 indicates that GluCla's are blocked at 5 μm and the results should declare this right at the start. The inference from Figure 2 as given in the fourth paragraph is “that GABAA receptors might play an additional role” is therefore unfounded by this data, whether it turns out to be true or not.

*Reviewer #3:*

"ON selectivity in the *Drosophila* visual system…" by Molina-Obando et al., is an excellent paper. In the paper, the authors show that ON-selectivity, via the L1 -> Mi1/Tm3 pathway is mediated by both glutamatergic and gaba-ergic ionotropic receptors. In addition, they show that the Mi1/Tm3 input must come from more than L1, i.e. that it is multi-synaptic. The authors use an impressive array of techniques for showing this, including pharmacology, genetics, electrophysiology, calcium imaging, etc, etc. I recommend the paper to be published, as it will be interesting to anyone working on ON selectivity, not only in the fly visual system, but also in vertebrates and other senses.

Subsection “The sign inversion in the ON pathway is a multisynaptic computation that depends on GluClα” second paragraph, you refer to Figure S3G, but I think you mean S4G. If so, you use it to show that L1 is glutamatergic, but the graph shows GABA-markers as well – how do you know that the GABA-ergic input cannot come from L1? Your summary diagrams all imply that GABA must come from something else.

You switch between PTX concentrations without being too clear as to why – maybe just refer to Figure 2 as justification? Furthermore, Figure 2B and D could be plotted as dose-response curves instead and you could extract IC50.

Figure 4C and similar (the same format used throughout), you have 3 graphs with different y-axes, but it is not entirely clear how these 3 parameters were extracted. Maybe you could add another panel where you describe the difference between step, plateau and integral?

Figure 5C,D, you need to explain what the little black squares and bars over the traces mean, and show how the parameters in panel E were extracted from these. The text mentions normalization to peak current, but this can be explained a lot clearer in the figure.

When you talk about your conclusions so far, you mention Rdl, but this is not in the diagram, whereas GluClalpha is clear in the text and diagram. I love these diagrams, btw, as they slowly add up. However, in Figure 6J you insert the unnamed element, which is not explained in the figure legend.

Figure 7A and 7G have identical pictograms, and similar y- and x-axes but different data. Please clarify. Figure 7D, "normalized to control", which is?

Figure 8, you could explain how the DSI and PD is extracted from an example trace, to clarify.

I don't understand the difference between Figure 3 and S3 – most of the information appears to be the same.

Figure legends, please separate the text clearly per panel, and not merged.

Figure 4 legend final sentence is very cumbersome. Consider breaking up.

---

## [Author Response]

Basically, the three reviewers were very positive about the paper and thought that it should be published. However, there are two important points that you should address before final acceptance of the manuscript:– One is the writing of the paper. The reviewers made many suggestions as to how to improve it to make it clearer for its audience. It is complex material and you should make every effort to help the readers. I recommend that you look at all the suggestions and come back with a version that addresses these points.– The other has to do with a point raised by reviewer #1 whether or not L1 cells co-express GABA and Glu. You should provide better evidence to support your point or simply be more careful with your statement on the lack of GABA release from L1 (e.g. subsection “A PTX insensitive GABAAR partially rescues ON responses”).Otherwise, congratulations on a very elegant paper that will have a significant impact in the field

First of all, we would like to thank you for the positive evaluation of our paper. We have followed your advice on how to improve many sections of the paper, and also worked with a professional editor to further improve the writing. We are especially grateful for your very detailed criticism. Thanks to that, we think that the manuscript is now an easier read and a stronger paper, and hope that you will agree.

Regarding the question if L1 neurons could release GABA, previous literature (e.g. Kolodziejczyk et al., 2008) showed that the only GABA-positive signal in the lamina comes from the axon terminals of C2 and C3 neurons, and not of any lamina monopolar cell. We also re-did the anti-GABA staining in a line that was specifically labeling L1 neurons, to demonstrate this more clearly, and again don’t see any GABA-staining in the L1 cell bodies, nor in the medulla neuropil containing the L1 terminals. L1 in fact does not express any major GABA synthesis enzymes, nor could GABA uptake from the extracellular milieu lead to GABA-release, because L1 also does not express the plasma-membrane GABA transporter Gat. VGaT alone (which is highly expressed in all neurons, as seen in Davis et al.) cannot account for GABA release, as this just accounts for vesicular uptake of GABA. We explain this more clearly in the paper

Reviewer #1:In this study, Molina-Obando and colleagues have investigated the cellular/circuit mechanisms underlying the emergence of ON selectivity, i.e., the specific sensitivity to increments of light intensity, within circuits of the fly visual medulla. Heretofore, these mechanisms have been poorly understood. Using a clever combination of pharmacology, genetics and physiological tools, this study reveals that: 1) inhibitory signals triggered by glutamate underlie signal transformation, 2) both Gabaergic and Glutamatergic synapses contribute to signal transformation, and 3) although there is a direct inhibitory glutamatergic synapse between presynaptic lamina neurons and the medulla cells under study, activity through this synapse is not critical for ON selectivity. Rather, signal transformation depends on a cascade of multiple synapses that recruit gabaergic- and glutamatergic-based inhibitory activity.Overall, the authors use a plethora of tools to address specific questions, which gives strength to the observations. However, in some cases, the description of the complex results is not very clear, is incomplete, or poorly discussed. This is important because the observations described in this study strongly suggest that early visual processing is organized in a distributed, multi-synaptic manner, expanding on the original idea of a monosynaptic functional organization. However, the interpretation of the complex results is not trivial, and in some cases, the lack of clarity in the text jeopardises the recognition of the value of the findings.I think this study is valuable not only for the fly vision community, but also for the broader sensory neuroscience community. The combination of pharmacology with genetic-based channel perturbations in a cell-type specific manner was key to demonstrate the distributed functional organization. Since the separation of signal processing in "ON" vs "OFF" pathways are found in many sensory systems across species, the implication of these results go beyond the fly vision community, and reaches the sensory neuroscience community at large. For these reasons, I would support publication of this work. However, as mentioned above, I found the manuscript a bit difficult to read.

We thank the reviewer for the very positive response, and the helpful criticism.

Reviewer #2:General assessment and major commentsIn this manuscript, the authors investigate the molecular mechanisms by which information is transmitted from the contrast encoding laminar monopolar cell L1 to medulla cells Mi1 and Tm3, two of the principal inputs to the direction-selective and ON motion-sensitive T4 cell. Their general strategy initially is to block channels using PTX and rescue function by expressing PTX resistant alleles. They find that low concentrations (5 μM) of PTX block both GABA_A and GluCla receptors (Figure 6). When PTX resistant Rdl alleles are expressed in Mi1 or Tm3, then the steady state response of these cells to ON contrast step changes is rescued (Figure 4). The authors then generate a PTX resistant GluCla allele, for which they are to be commended, and this rescues transient responses to ON contrast step changes (Figure 6). The Tm3 and Mi1 cells are situated in a network of cells including L5 and the GABAergic cells CT1 and C2, so it is entirely credible that L1 signals are polysynaptically communicated to Mi1 and Tm3 cells. In addition, they investigate the activity of T4 and T5 cells under PTX, and in flies with GluCla knocked out.Overall, I found the manuscript ended well with a great Discussion. The molecular work was impressive and clearly a lot of work. The organization of the results included a lot of implicit reasoning and odd ways of revealing important data that made it difficult to follow. Overall, I was not convinced that establishing a polsynaptic connection between L1 and Tm3 or Mi1 was surprising, and that the paper did not capitalize on this work to understand what the different mechanisms contributed to the ON rectification computation. I think that with a different emphasis, I might have been very impressed with the data. The generation of the PTX-insensitive GluCla receptor allele has the potential to be very useful tool for other circuit dissections. Mainly I have many minor comments, which reflect my enthusiasm for the work.

We thank the reviewer for the overall positive evaluation of our data. We worked on the writing, and especially added more explicit reasoning. We will discuss in the comments below our reasons to argue that we still think it is interesting that there are multisynaptic connections mediating the sign inversion in the ON pathway. However, we now also put a stronger focus on the pure mechanisms behind ON selectivity.

Major commentsThe first sentence of the second paragraph in subsection “ON responses are lost by PTX concentrations affecting GABAARs and GluCls” allow the reader to think that 100 μm PTX would block GABA_AR and GluCla and 5 μm PTX would block GABA_ARs alone. This allows one to think that GABA_Ars are required for normal glutamate signaling under 5 μm PTX in Figure 2. The data in Figure 6 indicates that GluCla's are blocked at 5 μm and the results should declare this right at the start. The inference from Figure 2 as given in the fourth paragraph is “that GABAA receptors might play an additional role” is therefore unfounded by this data, whether it turns out to be true or not.

Extensive previous literature shows that GluCls are generally blocked by higher concentrations of PTX than GABAARs. For instance, 100 μm PTX was previously used to block GluCls, whereas GABAARs are already blocked by lower concentrations of PTX (e.g. Liu and Wilson use 100 μm PTX to block GluCla and say “The concentration of picrotoxin we needed to achieve this level of blockade was higher than that needed to block GABA-gated chloride conductances in the same neurons”. Wilson and Laurent, 2005, state that “just 1 μm picrotoxin is sufficient to block 94.4% of the GABA-gated hyperpolarization in LNs”. In contrast, 21.1 μm PTX was needed to reduce the peak current amplitude of glutamate response of M. domestica GluCls to half (IC50), (Hirate et al., 2008). Furthermore, in this manuscript, it can be seen that 10 μm has a weaker effect on *D. melanogaster* GluCla than 100 μm PTX (oocyte data, Figure 5). Also, Fisher et al., 2015, used 5 μm PTX to block direction-selective responses in the visual system, and this effect could be rescued by the RdlMDRR insensitive allele, demonstrating specificity through the Rdl GABAAR in this context. Thus, the data shown in Figure 2 were precisely the data that suggested to us that “that GABAA receptors might play an additional role”, and lead us to further pursue the idea that GABAergic inhibition is also involved in mediating ON responses, which we confirmed by the use of the GABA insensitive RdlMDRR allele in Figure 4.

We are not sure what the reviewer means by “are required for normal glutamate signaling”. If the reviewer indeed meant glutamate signaling, we had shown that iGluSnFR signals are largely unaffected by 25 μm PTX (previous Figure S2). We now repeated these experiments at 5 μm and 100 μm PTX to better match other data in the paper and included them in a new Figure 2—figure supplement 2. We also added these data for Tm3 in Figure 2—figure supplement 3. If the reviewer instead meant “are required for normal ON responses”, then we would argue that this was exactly the objective that we were investigating here, and that these data thus allowed one to think that “GABAAR might play a role”.

We now expanded on the description of the previous literature on the different PTX concentrations used to make this clearer. “in vivostudies in *Drosophila* had previously used concentrations of 1-5 μM PTX to effectively block GABA-gated hyperpolarization in the olfactory system, and GABAergic inhibition in *Drosophila* visual system neurons (Wilson and Laurent, 2005, Fisher et al., 2015). In contrast, 100 μM PTX was used to block GluCls in the olfactory system (Liu and Wilson, 2013).”

Reviewer #3:"ON selectivity in the Drosophila visual system…" by Molina-Obando et al., is an excellent paper. In the paper, the authors show that ON-selectivity, via the L1 -> Mi1/Tm3 pathway is mediated by both glutamatergic and gaba-ergic ionotropic receptors. In addition, they show that the Mi1/Tm3 input must come from more than L1, i.e. that it is multi-synaptic. The authors use an impressive array of techniques for showing this, including pharmacology, genetics, electrophysiology, calcium imaging, etc, etc. I recommend the paper to be published, as it will be interesting to anyone working on ON selectivity, not only in the fly visual system, but also in vertebrates and other senses.Subsection “The sign inversion in the ON pathway is a multisynaptic computation that depends on GluClα” second paragraph, you refer to Figure S3G, but I think you mean S4G. If so, you use it to show that L1 is glutamatergic, but the graph shows GABA-markers as well – how do you know that the GABA-ergic input cannot come from L1? Your summary diagrams all imply that GABA must come from something else.

True, we fixed that figure reference and added a citation. We also added a detailed discussion of why L1 is likely not GABAergic. Please see our extensive response to reviewer #1 for this question, starting with “We apologize that we did not explain our argument that L1 is not GABAergic carefully enough,..”

You switch between PTX concentrations without being too clear as to why – maybe just refer to Figure 2 as justification? Furthermore, Figure 2B and D could be plotted as dose-response curves instead and you could extract IC50.

We now added a more detailed description of previous literature on why different PTX concentrations were thought to affect GABAAR and GluCls differentially (see our extensive answer to the major comment of reviewer #2). To avoid further confusions, we now also show data in Figure 2 at “low PTX” concentrations at 2.5 μM, to allow for direction comparisons with Figure 4 and 6. We also added sentences at several places in the Results section, explicitly introducing why we did an experiment at a given PTX concentration.

We cannot really derive IC50, because PTX is unspecific, and we would measure confounding effects on GABAAR and GluCls. (Actually, when you squint at the Tm3 data in Figure 2, you might see that the PTX effect doesn’t appear to increase continuously, but rather step wise).

Figure 4C and similar (the same format used throughout), you have 3 graphs with different y-axes, but it is not entirely clear how these 3 parameters were extracted. Maybe you could add another panel where you describe the difference between step, plateau and integral?

We have added a schematic illustrating these quantifications to Figure 3—figure supplement 1, mention the following sentence in the Results section: “We individually quantified the amplitude of the maximum response to the ON step, the amplitude of the plateau response, and the integrated response during the ON step (Figure 4—figure supplement 1A).” and expanded the description of the analysis in the Materials and methods section.

Figure 5C,D, you need to explain what the little black squares and bars over the traces mean, and show how the parameters in panel E were extracted from these. The text mentions normalization to peak current, but this can be explained a lot clearer in the figure.

We now explain this better in the figure legend

When you talk about your conclusions so far, you mention Rdl, but this is not in the diagram, whereas GluClalpha is clear in the text and diagram. I love these diagrams, btw, as they slowly add up. However, in Figure 6J you insert the unnamed element, which is not explained in the figure legend.

We now write Rdl (GABA-A-R)

Figure 7A and 7G have identical pictograms, and similar y- and x-axes but different data. Please clarify.

In Figure 7A, the genome of the fly carries the FlpStop exon in the non-disrupting (ND) orientation, and is then inverted (by Flp recombinase) into the disrupting orientation in all neurons (elav>>Flp). This test is done in a heterozygous background (xxx **/ +**). Therefore, this is testing for both the efficiency of FlpStop and the efficiency of inversion. It is almost as good as it could be (50% mRNA levels).

In Figure 7G, the genome of the fly carries the FlpStop exon in the disrupting orientation. Here, we tested the heterozygous condition (FlpStop.D/+), which should be similar to elav>>Flp. A small discrepancy between panels A and G can potentially be accounted for by cells expressing GluCl, but not expressing elav>>Flp, or by non-complete inversion. Here, we also tested the full, homozygous mutant, by having the GluCla-FlpStop allele in trans to a deficiency. Strikingly, there is almost no mRNA in this condition.

We added a sentence to explain the latter case better: “To test this, we used a FlpStop allele inserted in the disrupting orientation. In this background, expression should be fully disrupted in all cells normally expressing GluClα.”

Figure 7D, "normalized to control", which is?

The controls were wild type flies. We replaced the x-axis label with “*wild type (+/+)*” to make that clearer

Figure 8, you could explain how the DSI and PD is extracted from an example trace, to clarify.

We defined PD as the direction of motion that elicited the strongest responses. The Materials and methods section says: “The direction selectivity index (DSI) was calculated as PD-ND/PD, where PD is the maximum response among all responses to the different directions of motion and ND is the response to the null direction of motion defined as 180° from the preferred direction.”

Because our main figure showed the mean of all traces (which could have different individual PDs), we could add a supplementary figure that illustrates this for an example fly (Author response image 1), but we almost feel that it’s too standard to do this.

**Author response image 1. respfig1:** Extraction of PD and ND responses from T4/T5 calcium imaging data. (**A**) Expression of GCaMP6f in T4/T5 neurons. ROIs are manually drawn and assigned to different lobula plate (LP) or medulla (Me) layers. (**B**) Single ROI responses to ON (bright background) and OFF (grey background) moving into different directions of motion. (**C**) Upon averaging across trials and ROIs, the preferred direction (PD) is determined as the direction of motion that was eliciting the strongest response. The null direction (ND) is set as the direction of motion that is 180° relative to the PD.

I don't understand the difference between Figure 3 and S3 – most of the information appears to be the same.

Two separate studies performed cell-type-specific RNAseq of the adult *Drosophila* visual system. We showed the Davis et al. 2018 data in the main figure because this is more comprehensive for the cell types we were interested in. At the same time, we also wanted to give credit to the data from the Desplan lab (Konstantinides et al., 2018) and also thought it is good to see how much these two studies overlap. We therefore show those data in the Supplements.

Figure legends, please separate the text clearly per panel, and not merged.

Done.

Figure 4 legend final sentence is very cumbersome. Consider breaking up.

Done.